# PRODuctive bandits: Importance Weighting No More

**Julian Zimmert**
Google Research
zimmert@google.com

**Teodor V. Marinov**
Google Research
tvmarinov@google.com

## Abstract

Prod is a seminal algorithm in full-information online learning, which has been conjectured to be fundamentally sub-optimal for multi-armed bandits. By leveraging the interpretation of Prod as a first-order OMD approximation, we present the following surprising results: 1. Variants of Prod can obtain optimal regret for adversarial multi-armed bandits. 2. There exists a simple and (arguably) importance-weighting free variant with optimal rate. 3. One can even achieve best-both-worlds guarantees with logarithmic regret in the stochastic regime.

The bandit algorithms in this work use simple arithmetic update rules without the need of solving optimization problems typical in prior work. Finally, the results directly improve the state of the art of incentive-compatible bandits.

## 1 Introduction

The adversarial multi-armed bandit (MAB) problem is a seminal online learning problem with applications in experimental design, online advertisement and more [Thompson, 1933, Lai and Robbins, 1985, Auer et al., 2002a,b]. MABs are characterized by the limited feedback given to the learner in every round, the so-called bandit feedback, in which the learner only observes the loss of their selected action, unlike in the full information, also known as the experts, setting where the loss of all actions are provided as feedback.

The first nearly optimal algorithm for the adversarial MAB problem is EXP3 [Auer et al., 2002b]. EXP3 is a direct adaptation of the Hedge algorithm [Littlestone and Warmuth, 1994, Cesa-Bianchi et al., 1997, Freund and Schapire, 1997] with importance weighting to handle partial bandit feedback. Hedge and EXP3 are special versions of online mirror descent (OMD), where the Bregman divergence is the KL divergence induced by the Negative entropy potential. The OMD view of online learning [Abernethy et al., 2008] has lead to a wide range of MAB algorithms such as Tsallis-INF [Audibert and Bubeck, 2009] and Logbarrier [Agarwal et al., 2017], which enjoy improved regret guarantees. These guarantees can be attributed to regularizers more suited to the bandit feedback setting, compared to the negative entropy regularizer. A downside of OMD is that usually the mirror descent update is not closed form and requires (approximately) solving optimization problems at every iteration.

Alternative full-information algorithms with simple arithmetic updates are Prod [Cesa-Bianchi et al., 2007, Even-Dar et al., 2008, Gaillard et al., 2014], which enjoys second order regret bounds and Multiplicative Weights Update (MWU) [Arora et al., 2012]. Prod is known to be closely related to Hedge, both of which are different generalizations of the weighted majority algorithm [Littlestone and Warmuth, 1994] to non-binary feedback.

More recently, Freeman et al. [2020] studied incentive-compatible online learning, a setting where experts are not necessarily truthful but make predictions strategically with regard to the agent's algorithm. Motivated by deriving an algorithm where the incentives of experts align with the agent,

38th Conference on Neural Information Processing Systems (NeurIPS 2024).

they propose WSU which can be seen as an instance of Prod. Freeman et al. [2020] further introduce a bandit adaptation WSU-UX, the first Prod algorithm for bandits. Unfortunately, WSU-UX only enjoys $T^{\frac{2}{3}}$ regret guarantees. This is not merely an issue with their analysis as has been shown via lower bounds [Mortazavi et al., 2024] and leads to the conjecture that this might be a fundamental separation between full-information and bandits.

We make the following contributions for understanding Prod under bandit feedback:

1. We disprove the separation conjecture by providing a simple modification of WSU-UX with nearly optimal $O(\sqrt{KT\log(K)})$ regret guarantees.

2. We present a Prod variant that does not require importance weighting and yet enjoys $O(\sqrt{KT\log(T)})$ regret bounds.

3. We present a Prod variant that achieves best of both worlds regret guarantees, i.e. it enjoys improved $O(\log(T))$ regret bounds when the losses are stochastic, while maintaining worst-case $O(\sqrt{KT})$ regret bounds.

**Notation:** For $N \in \mathbb{N}$, let $[N] = \{1, \ldots, N\}$. We use $\langle \cdot, \cdot \rangle$ to denote the regular Euclidean scalar product and $\Delta(A)$ to denote the probability simplex over a finite set $A$. $O$ is the standard Landau notation hiding numerical constants, while $\widetilde{O}$ omits polylogaritmic factors as well. The expectation $\mathbb{E}$ is always taken over all randomness of the algorithm, losses and experts, while $\mathbb{E}_t[\cdot] = \mathbb{E}[\cdot | \mathcal{F}_t]$, where $\mathcal{F}_t$ is the filtration over all randomness up to step $t$. $\mathbb{I}(E)$ denotes the indicator function function for the event $E$. For a convex differentiable function $F$, the Bregman divergence is defined by $D_F(y, x) = F(y) - F(x) - \langle y - x, \nabla F(x) \rangle$.

## 2 Problem setting and related work

The adversarial bandit problem is formally defined as follows. In every round $t = 1, \ldots, T$, an (oblivious) adversary selects a loss $\ell_t \in [0,1]^K$ (it is possible to extend the loss range to $[-1,1]^K$) unknown to the agent. The agent simultaneously selects an expert $A_t \sim \pi_t$, $\pi_t \in \Delta([K])$. The agent incurs and observes the loss $\ell_{t,A_t}$, but does not see the losses of other experts. The goal is to minimize the pseudo-regret[1]

$$\text{Reg} = \max_{i \in [K]} \mathbb{E}\left[\sum_{t=1}^{T} \ell_{t,A_t} - \ell_{t,i}\right] = \max_{u \in \Delta([K])} \mathbb{E}\left[\sum_{t=1}^{T} \langle \pi_t - u, \ell_t \rangle\right],$$

Popular families of algorithms for this problem setting include online mirror descent (OMD) and follow the regularized leader (FTRL). Typically the algorithms use unbiased loss estimates of the loss vector via importance weighting: $\hat{\ell}_{t,i} = \frac{\ell_{t,i}}{\pi_{t,i}}\mathbb{I}(A_t = i)$. We note that other types of importance weighted estimators have been used in literature such as the implicit exploration estimator Kocák et al. [2014], which has improved variance properties. The algorithms are defined by a twice-differentiable convex potential function $F : \mathbb{R}^K \to \mathbb{R}$ and a learning rate schedule $\eta_t$, the agent maintains a distribution via

$$\pi_{t+1} = \arg\min_{\pi \in \Delta([K])} \left\langle \pi, \eta_t \hat{\ell}_t \right\rangle - D_F(\pi, \pi_t), \tag{OMD}$$

$$\pi_{t+1} = \arg\min_{\pi \in \Delta([K])} \left\langle \pi, \eta_t \sum_{s=1}^{t} \hat{\ell}_s \right\rangle - F(\pi), \tag{FTRL}$$

OMD optimizes locally given the last loss and, as we will show, is most closely related to Prod. FTRL on the other hand performs a global optimization and is generally considered superior for adaptive bounds with time-dependent learning rates. In some special cases, such as time-independent learning rate with potentials that satisfy $\|\nabla F(x)\| \to \infty$ on the border of the optimization set, both algorithms are equivalent. Common potentials in the bandit literature are given in Table 1 and we refer to their respective Bregman divergences as $D_{KL}, D_{TS}$ and $D_{LB}$ respectively. The negative entropy is the potential which defines Hedge and Exp-3 (and derivatives) [Littlestone and

---

[1]For the rest of the paper we refer to pseudo-regret as regret for simplicity.

| $F(\pi)$ | Negentropy/KL divergence | 1/2-Tsallis Entropy | Logbarrier |
|---|---|---|---|
| | $\sum_{i=1}^{K} \pi_{t,i} \log(\pi_{t,i})$ | $-2\sum_{i=1}^{K} \sqrt{\pi_{t,i}}$ | $-\sum_{i=1}^{K} \log(\pi_{t,i})$ |

Table 1: Common potential functions

Warmuth, 1994, Vovk, 1995, Freund and Schapire, 1997, Auer et al., 2002b, Kocák et al., 2014]. The 1/2-Tsallis Entropy is the key to achieving optimal best-of-both-worlds regret guarantees as was first demonstrated by Zimmert and Seldin [2021]. The Logbarrier potential was used by Agarwal et al. [2017] to first solve the corralling of bandits problem and has found many applications in model-selection problems [Foster et al., 2020], regret bounds which depend on the properties of the loss sequence [Wei and Luo, 2018, Lee et al., 2020b,a] and various other bandit problems.

## 2.1 Prod family of algorithms

The original version of Prod [Cesa-Bianchi et al., 2007] maintains weights $w_{t,i}$ for each experts which are updated via $w_{t+1,i} = w_{t,i}(1 - \eta\ell_{t,i})$ and the agent plays the policy $\pi_{t,i} \propto w_{t,i}$. This framework has been extended to D-Prod [Even-Dar et al., 2008], which shifts the losses in the weight update by the loss of a fixed policy, and ML-Prod [Gaillard et al., 2014] that shifts losses by the mean of the current policy (among other modifications). With a suitable shift in losses, one can ensure that the weights sum up to 1 and hence operate directly on the policy space. In its simplest form, this is

$$\pi_{1,i} = \frac{1}{K}, \qquad \pi_{t+1,i} = \pi_{t,i}(1 - \eta(\ell_{t,i} - \lambda_t)), \qquad \lambda_t = \sum_{j=1}^{K} \pi_{t,j}\ell_{t,j}. \quad \text{(Vanilla-Prod/WSU)}$$

We refer to this update as Vanilla-Prod to emphasize its connection to the Prod literature, however this algorithm is exactly WSU [Freeman et al., 2020] derived for incentive-compatible online learning. We consider any algorithm a variant of Prod if it performs product updates of the form $\pi_{t+1,i} = \pi_{t,i}(1 - \eta L_{t,i}(\ell_t; A_t))$, where $L_{t,i}$ are linear affine functions of the loss. From now on we always assume the initial policy is $\pi_{t,i} = 1/K$, and this holds for all algorithms presented in the paper. The appeal of Prod algorithms lies in their simple arithmetic update rule. A second motivation for using Prod updates is the mentioned incentive-compatibility.

## 2.2 Incentive-compatible online learning

In the incentive-compatible online learning setting, introduced by Freeman et al. [2020], experts provide recommendations, for example a prediction of whether it will rain on the next day. Each expert has an internal belief and the agent would like to receive each expert's true beliefs in order to learn to follow the best expert. In the simplest setting the experts make predictions about binary outcomes, with the $i$-th expert having (private) belief $b_{t,i} \in [0,1]$ about the $t$-th round outcome. The expert's belief is unknown to the agent and the expert only reports a prediction $p_{t,i} \in [0,1]$ about the outcome. Based on the expert predictions, $\{p_{t,i}\}_{i \in [K]}$ the agent makes a prediction based on $\bar{p}_t = \sum_{i=1}^{K} \pi_{t,i} p_{t,i}$ and incurs a loss $\mathcal{L}(\bar{p}_t, r_t) \in [0,1]$ based on the realized outcome $r_t \in \{0,1\}$. In the weather forecasting example the outcome is the indicator if it rains the next day and the loss is $\mathcal{L}(\bar{p}_t, r_t) = (r_t - \bar{p}_t)^2$. In Freeman et al. [2020] the experts only care about maximizing the probability that they are selected which does not necessarily result in truthful reporting, that is $b_{t,i}$ may differ from $p_{t,i}$. The agent's goal of receiving the true beliefs, $\{b_{t,i}\}_{i \in [K]}$, can be achieved by playing an *incentive compatible strategy* which will always prefer selecting an truthful expert, that is the probability of $\pi_{t+1,i}$ of selecting expert $i$ when the expert reports $b_{t,i}$ may only decrease if the expert reports any other $p_{t,i}$ instead, no matter how the remaining experts act throughout the game. This is made precise in Definition 2.1 of Freeman et al. [2020].

Freeman et al. [2020] show that standard OMD and FTRL algorithms are in fact not incentive compatible even when the loss $\mathcal{L}$ is restricted to be *proper* that is $\mathbb{E}_{r \sim \text{Bern}(b)}[\mathcal{L}(p,r)] \geq \mathbb{E}_{r \sim \text{Bern}(b)}[\mathcal{L}(b,r)]$ for all $p \neq b$. It turns out that any update for $\pi_{t+1}$ which is linear affine in the proper loss function will lead to incentive compatibility and so the Prod family will ensures that experts report their true believes in this setting, i.e. they are incentive-compatible. The state of the art for incentive-compatible bandits is $T^{\frac{2}{3}}$ regret and any improvement for Prod directly transfers to better rates for this setting as well.

## 3 Modifying WSU-UX for nearly optimal regret guarantees

We begin by presenting a minimal modification of Algorithm WSU-UX which is sufficient for a regret guarantee of the order $O(\sqrt{KT\log(K)})$.

WSU-UX uses importance-weighted updates and injects a small uniform exploration.

$$\tilde{\pi}_{t,i} = \frac{\gamma}{K} + (1-\gamma)\pi_{t,i}, \qquad A_t \sim \tilde{\pi}_{t,i}, \qquad \hat{\ell}_{t,i} = \mathbb{I}(A_t = i)\frac{\ell_{t,i}}{\tilde{\pi}_{t,i}}$$

$$\pi_{t+1,i} = \pi_{t,i}(1 - \eta(\hat{\ell}_{t,i} - \lambda_t)), \qquad \lambda_t = \sum_{j=1}^{K}\pi_{t,j}\hat{\ell}_{t,j}, \qquad \text{(WSU-UX)}$$

where $\gamma$ is the mixture coefficient. The role of uniform exploration is to ensure that the policy updates are proper i.e. $\pi_{t+1,i} \in (0,1)$. Freeman et al. [2020] uses the following key lemmas in their analysis, which hold for any sequence of losses $\ell_t \in [0,1]^K$. For completeness we restate the results we use below.

**Lemma 1** (Lemma 4.1 [Freeman et al., 2020]). *If $\eta K/\gamma \leq \frac{1}{2}$, the WSU-UX weights $\pi_t$ and $\tilde{\pi}_t$ are valid probability distributions for all $t \in [T]$.*

**Lemma 2** (Lemma 4.3 [Freeman et al., 2020]). *For WSU-UX, the probability vectors $\{\pi_t\}_{t\in[T]}$ and loss estimators $\hat{\ell}_t$ satisfy the following second order-bound*

$$\sum_{t=1}^{T}\sum_{i=1}^{K}\pi_{t,i}\hat{\ell}_{t,i} - \sum_{t=1}^{T}\hat{\ell}_{t,i^\star} \leq \frac{\log(K)}{\eta} + \eta\sum_{t=1}^{T}\hat{\ell}_{t,i^\star}^2 + \eta\sum_{t=1}^{T}\sum_{i=1}^{K}\pi_{t,i}\hat{\ell}_{t,i}^2,$$

*where $i^\star$ is the optimal expert/arm.*

The bound in Lemma 2 is almost the standard regret bound that appears in the analysis of Hedge, except for the term $\eta\sum_{t=1}^{T}\hat{\ell}_{t,i^\star}^2$. This term is the reason why prior work can not show regret bounds smaller than $T^{\frac{2}{3}}$. Even after taking the expectation over the randomness of the agents actions, this term scales with with $1/\pi_{t,i^\star}$, which is potentially unbounded. Alternatively this term can be written as $\mathbb{E}_t^{j\sim\pi^\star}[\eta\hat{\ell}_{tj}^2]$ (where $\pi^\star$ is the policy picking $i^\star$ with probability 1) and if one could perform a change of measure to $\mathbb{E}_t^{j\sim\tilde{\pi}_t}[\eta\hat{\ell}_{tj}^2]$, this term is immediately controllable.

In fact, change of measure techniques for bandits are now well established [Foster et al., 2020, Luo et al., 2021] by introducing biases to the losses. Assume we construct a bias to the losses $\tilde{\ell}_t = \ell_t + \delta_t$, which satisfies the same regret guarantee, Reg, as the original loss sequence, then running an algorithm over the biased loss sequence $\tilde{\ell}_t$ which selects $A_t \sim \tilde{\pi}_t$ ensures

$$\mathbb{E}\left[\sum_{t=1}^{T}\ell_{t,A_t} - \ell_{t,i^\star}\right] = \mathbb{E}\left[\sum_{t=1}^{T}\tilde{\ell}_{t,A_t} - \tilde{\ell}_{t,i^\star} + \delta_{t,A_t} - \delta_{t,i^\star}\right]$$

$$= \text{Reg} + \mathbb{E}\left[\sum_{t=1}^{T}\underbrace{(\mathbb{E}_t^{j\sim\tilde{\pi}_t}[\delta_{t,j}] - \mathbb{E}_t^{j\sim\pi^\star}[\delta_{t,j}])}_{\text{change of measure}}\right].$$

We introduce now the following modification to the losses

$$\tilde{\ell}_{t,i} = \ell_{t,i}\left(1 - \frac{\eta}{\tilde{\pi}_{t,i}}\right), \qquad \hat{\ell}_{t,i} = \mathbb{I}(A_t = i)\frac{\tilde{\ell}_{t,i}}{\tilde{\pi}_{t,i}}. \qquad (1)$$

which corresponds to $\delta_{ti} = \frac{\eta\ell_{ti}}{\tilde{\pi}_{t,i}}$. This yields the change of measure term

$$\mathbb{E}_t^{j\sim\tilde{\pi}_t}[\delta_{t,j}] - \mathbb{E}_t^{j\sim\pi^\star}[\delta_{t,j}] = \sum_{i=1}^{K}\eta\ell_{ti} - \mathbb{E}_t^{j\sim\pi^\star}\left[\frac{\eta\ell_{tj}}{\tilde{\pi}_{tj}}\right] \leq \eta K - \mathbb{E}_t^{j\sim\pi^\star}[\eta\hat{\ell}_{tj}^2],$$

which is sufficient for controlling the term $\eta\sum_{t=1}^{T}\hat{\ell}_{t,i^\star}^2$ in Lemma 2.

**Theorem 1.** *Running WSU-UX with the loss estimators in Equation 1 and $\gamma = \frac{\eta K}{2}, \eta = \Theta(\sqrt{\frac{\log(K)}{KT}})$ guarantees the following regret bound*

$$\sum_{t=1}^{T} \mathbb{E}[\ell_{t,A_t} - \ell_{t,i^\star}] \leq O(\sqrt{KT \log(K)}).$$

The proof of Theorem 1 is deferred to Appendix B.

## 3.1 Intuition on biasing the update and the Prod family of algorithms

We provide an intuition in this section about why WSU-UX is not tight and why the bias we choose is able to correct the regret. The modern analysis of OMD (or FTRL) with a divergence function $D$ is follows the template[2]

$$\sum_{t=1}^{T} \langle \pi_t^{\text{OMD}} - \pi^\star, \ell_t \rangle = \sum_{t=1}^{T} \left( \langle \pi_t^{\text{OMD}} - \pi^\star, \ell_t \rangle + \frac{D(\pi^\star, \pi_{t+1}^{\text{OMD}}) - D(\pi^\star, \pi_t^{\text{OMD}})}{\eta} \right)$$

$$+ \frac{D(\pi^\star, \pi_1^{\text{OMD}}) - D(\pi^\star, \pi_{T+1}^{\text{OMD}})}{\eta} \leq \sum_{t=1}^{T} \left( \underbrace{\langle \pi_t^{\text{OMD}} - \pi_{t+1}^{\text{OMD}}, \ell_t \rangle + \frac{D(\pi_{t+1}^{\text{OMD}}, \pi_t^{\text{OMD}})}{\eta}}_{\text{stability}} \right) + \frac{D(\pi^\star, \pi_1^{\text{OMD}})}{\eta},$$

(2)

where the inequality crucially relies on $\pi_{t+1}^{\text{OMD}}$ being the 1-step OMD update to the previous policy. OMD algorithms like Hedge choose the policy that minimizes the per-step *stability* term in every round, which is what allows for the stability term to be bounded appropriately. If instead of playing $\pi_t^{\text{OMD}}$ an approximate policy $\pi_t \approx \pi_t^{\text{OMD}}$ is played, the template regret analysis can be changed by adding the terms

$$\eta^{-1}(D(\pi^\star, \pi_{t+1}) - D(\pi^\star, \pi_{t+1}^{\text{OMD}})),$$

where $\pi_{t+1}^{\text{OMD}}$ is now the 1-step OMD update from $\pi_t$. When $D$ is the KL divergence and the approximate policy $\pi_t$ is coming from the WSU-UX, this term contributes the undesirable $\eta \hat{\ell}_{t,i^\star}^2$.

We now explain how our loss biasing solves this issue. The Vanilla-Prod/WSU update can be seen as a first order approximation to the Hedge update, that is

$$\pi_{t+1,i}^{\text{OMD}} = \pi_{t,i} \exp(-\eta(\hat{\ell}_{t,i} - \lambda_t)) \underbrace{\approx}_{\text{first-order}} \pi_{t,i}(1 - \eta(\hat{\ell}_{t,i} - \lambda_t)) = \pi_{t,i},$$

where $\lambda_t$ is a normalization factor. Tuning $\lambda_t$ such that $\sum_{i=1}^{K} \pi_{t+1,i} = 1$ recovers Vanilla-Prod/WSU. To control the undesirable terms, we have to make the approximation tighter. The loss-biasing introduced in the previous section acts as a correction which brings the Vanilla-Prod/WSU update closer to the second order approximation of the Hedge update. Indeed, we have

$$\pi_{t+1,i}^{\text{OMD}} = \pi_{t,i} \exp(-\eta(\hat{\ell}_{t,i} - \lambda_t)) \underbrace{\approx}_{\text{second-order}} \pi_{t,i} \left( 1 - \eta(\hat{\ell}_{t,i} - \lambda_t) + \frac{\eta^2}{2}(\hat{\ell}_{t,i} - \lambda_t)^2 \right)$$

$$= \pi_{t,i} \left( 1 - \eta(\hat{\ell}_{t,i} - \lambda_t) \left( 1 - \frac{\eta}{2}(\hat{\ell}_{t,i} - \lambda_t) \right) \right),$$

and so our loss adjustment in Equation 1, $\tilde{\ell}_{t,i} = \ell_{t,i}(1 - \eta/\tilde{\pi}_{t,i})$, can be seen as a second order correction to the term $\frac{\eta^2}{2}(\hat{\ell}_{t,i} - \lambda_t)^2$. We cannot exactly correct the second order difference with linear update rules, which we address by slightly overcorrecting, i.e. biasing by a larger amount than the second order adjustments implies as necessary. That is, the correction term we use is of the order $\eta/\tilde{\pi}_{t,i}$ instead of $\eta\ell_{t,i}/\tilde{\pi}_{t,i}$. Fortunately, the regret analysis is not sensitive towards this as we have shown in Theorem 1.

---

[2]This might look very dissimilar from the original Hedge/EXP/MWU analysis, but it is actually equivalent after accounting for the special form of the KL divergence.

# 4 Importance weighting free adversarial MAB with LB-Prod

While our biased WSU-UX obtains optimal regret, it still has to go through the extra complexity of injecting additional uniform exploration at a rate of $\gamma$ to ensure proper updates and add bias to the losses. As mentioned in the introduction, prior work proposed other potential functions that have favourable properties for bandit feedback. Using the same linearization argument to derive a Prod version based on the Logbarrier leads to a surprisingly simple algorithm without loss biasing that is arguably importance weighting free. LB-Prod differs from WSU-UX by using the masked loss $\tilde{\ell}_{t,i} = \ell_{t,i}\mathbb{I}(A_t = i)$ instead of the importance weighted loss and a non-symmetric normalization $\lambda_{t,i}$:

$$\pi_{t+1,i} = \pi_{t,i}(1 - \eta(\tilde{\ell}_{t,i} - \lambda_{t,i})), \qquad \lambda_{t,i} = \pi_{t,i}\frac{\pi_{t,A_t}\ell_{t,A_t}}{\sum_{j=1}^{K}\pi_{t,j}^2}. \qquad \text{(LB-Prod)}$$

It is easy to confirm $\tilde{\ell}_{t,i} - \lambda_{t,i} \in [-1,1]$ via $\pi_{t,i}\pi_{t,A_t} \leq (\pi_{t,i}^2 + \pi_{t,A_t}^2)/2$ yielding proper updates for $\eta < 1$. The following theorem shows that this simple algorithm is rate optimal under the right tuning.

**Theorem 2.** *For any sequence of losses $\ell_t \in [-1,1]^K$ and any $\eta < 1$, LB-Prod produces valid distributions $\pi_t \in \Delta([K])$ and its regret is bounded by*

$$\sum_{t=1}^{T}\mathbb{E}[\ell_{t,A_t} - \ell_{t,i^\star}] \leq 2 + \frac{K\log(T)}{\eta} + \frac{2\eta T}{1-\eta}.$$

Tuning $\eta = \sqrt{\frac{\log(T)K}{2T}}$ results in a regret bound of $O(\sqrt{KT\log(T)})$ for any $T > \frac{K\log(T)}{2}$. The proof of Theorem 2 is deferred to the end of the section.

## 4.1 Intuition of LB-Prod

As mentioned before, LB-Prod is the linear approximation of OMD with Bregman divergence induced by the Logbarrier potential, that is $D_{LB}$ induced by the potential fuction $F(x) = -\sum_{i=1}^{K}\log(x_i)$. The one-step Logbarrier OMD update of a policy $\pi_t$ with importance-weighting is known (see e.g. [Zimmert and Seldin, 2021]) to take the form

$$\pi_{t+1,i}^{\text{OMD}} = \frac{\pi_{t,i}}{1 + \eta\pi_{t,i}(\hat{\ell}_{t,i} - \lambda_t)} \approx \pi_{t,i}(1 - \eta\pi_{t,i}(\hat{\ell}_{t,i} - \lambda_t)) = \pi_{t+1,i},$$

where $\lambda_t$ is a normalization constant that ensures $\pi_t^{\text{OMD}}$ is a probability distribution. If instead $\lambda_t$ is tuned so that $\sum_{i=1}^{K}\pi_{t+1,i} = 1$, the LB-Prod update is recovered. This can be be verified by setting $\tilde{\ell}_{t,i} = \pi_{t,i}\hat{\ell}_{t,i}$ and $\lambda_{t,i} = \pi_{t,i}\lambda_t$. The curvature of the Logbarrier regularization is what ensures that the importance weighted loss $\hat{\ell}_{t,i}$ is always multiplied with its probability $\pi_{t,i}$, allowing to run the algorithm on the masked non-weighted loss sequence directly.

Additionally, the second order approximation is

$$\pi_{t+1,i}^{\text{OMD}} = \frac{\pi_{t,i}}{1 + \eta\pi_{t,i}(\hat{\ell}_{t,i} - \lambda_t)} \underbrace{\approx}_{\text{second-order}} \pi_{t,i}(1 - \eta\pi_{t,i}(\hat{\ell}_{t,i} - \lambda_t) + \eta^2\pi_{t,i}^2(\hat{\ell}_{t,i} - \lambda_t)^2).$$

The additional undesirable terms, unlike in the case for WSU-UX, will only contribute $\eta T$ regret if not adjusted for as we show next.

## 4.2 Analysis of LB-Prod

The following technical lemma is proven in the appendix.

**Lemma 3.** *For any timestep $t$ and arm $i$, it holds*

$$\mathbb{E}_t[\tilde{\ell}_{t,i} - \lambda_{t,i}] = \pi_{t,i}(\ell_{t,i} - c_t), \qquad \mathbb{E}_t[(\tilde{\ell}_{t,i} - \lambda_{t,i})^2] \leq 2\pi_{t,i},$$

*where $c_t \in [-1,1]$ is an arm independent constant.*

In Section 3.1 we argued that one needs to bound the additional term $\eta^{-1}(D_{LB}(\pi^\star, \pi_{t+1}) - D_{LB}(\pi^\star, \pi_{t+1}^{\text{OMD}}))$ to reduce the analysis to standard OMD. While this term is nicely bounded for LB-Prod, it turns out that it is easier to directly bound the "prototype" of the *stability* term in Equation (2) due to the fact that we have a closed form expression of $\pi_{t+1}$.

**Lemma 4.** *For any time $t \in [T]$ and any $u \in \Delta([K])$, it holds*

$$\langle \pi_t - u, \ell_t \rangle + \mathbb{E}_t \left[ \eta^{-1} D_{LB}(u, \pi_{t+1}) \right] - \eta^{-1} D_{LB}(u, \pi_t) \leq \frac{2\eta}{1 - \eta} \,.$$

The proof is an algebraic exercise and deferred to the supplementary material. Finally, we can prove the main regret guarantee.

*Proof of Theorem 2.* To show that this algorithm outputs proper probability distributions, note that

$$\sum_{i=1}^{K} \pi_{t+1,i} = \left( \sum_{i=1}^{K} \pi_{t,i} \right) - \eta \pi_{t,A_t} \ell_{t,A_t} + \eta \sum_{j=1}^{K} \frac{\pi_{t,A_t} \pi_{t,j}^2}{\sum_{k=1}^{K} \pi_{t,k}^2} \ell_{t,A_t} = \sum_{i=1}^{K} \pi_{t,i} = \cdots = \sum_{i=1}^{K} \pi_{1,i} = 1 \,.$$

Additionally we have seen that $|\tilde{\ell}_{t,i} - \lambda_{t,i}| \leq 1$, hence for any $\eta < 1$, the probability of any arm is strictly positive. For any comparator $u^\star$, we define $u = u^\star + \frac{1}{T}(\pi_1 - u^\star)$, which satisfies $\sum_{t=1}^{T} \langle u - u^\star, \ell_t \rangle \leq 2$. Using Lemma 4 and Equation (2), we obtain by the telescoping sum of Bregman terms

$$\mathbb{E} \left[ \sum_{t=1}^{T} \langle \pi_t - u, \ell_t \rangle \right] \leq \frac{4\eta T}{1 - \eta} + \eta^{-1} \mathbb{E} \left[ D_{LB}(u, \pi_1) - D_{LB}(u, \pi_{T+1}) \right] \leq \frac{4\eta T}{1 - \eta} + \frac{K \log(T)}{\eta} \,.$$

$\square$

### 4.3 The perturbation analysis

We outline an alternative analysis that reuses established machinery and might be more accessible for some readers. Our analysis begins by viewing the Prod update as an exact OMD update over a perturbed loss sequence. Indeed, there is a sequence of perturbations $\{\epsilon_t\}_{t \in [T]}, \epsilon_t \in \mathbb{R}^K$, such that

$$\pi_{t+1}^{\text{OMD}} = \frac{\pi_{t,i}}{1 + \eta \pi_{t,i}(\hat{\ell}_{t,i} - \epsilon_{t,i} - \lambda_t)} = \pi_{t,i}(1 - \eta \pi_{t,i}(\hat{\ell}_{t,i} - \lambda_t)) = \pi_{t+1,i} \,.$$

The exact form of $\epsilon_{t,i}$ satisfies the following

$$\epsilon_{t,i} = \frac{\eta \pi_{t,i}(\hat{\ell}_{t,i} - \lambda_t)^2}{1 + \eta \pi_{t,i}(\hat{\ell}_{t,i} - \lambda_t)} \,, \qquad |\mathbb{E}_t[\epsilon_{t,i}]| = O(\eta) \,.$$

Since Prod is exactly OMD over the sequence $\hat{\ell}_t - \epsilon_t$, we can decompose the regret as follows

$$\mathbb{E} \left[ \sum_{t=1}^{T} \left\langle \pi_t - u, \hat{\ell}_t \right\rangle \right] = \mathbb{E} \left[ \sum_{t=1}^{T} \left\langle \pi_t - u, \hat{\ell}_t - \epsilon_t \right\rangle \right] + \mathbb{E} \left[ \sum_{t=1}^{T} \langle \pi_t - u, \epsilon_t \rangle \right] = \text{Reg}_{\text{OMD}} + O(\eta T) \,.$$

The analysis is not entirely straightforward as the loss range for the OMD update becomes $[-1 - O(\eta), 1 + O(\eta)]$ because of the shift introduced by the perturbation of the losses, and this posses some additional difficulties.

## 5 Best of both worlds algorithms

In applications where the loss is potentially more benign, for example sampled i.i.d. from a a fixed distribution over $[0,1]^K$, it is desirable to obtain faster rates in nice environments while preserving worst-case guarantees. Probably the simplest algorithm with this property is Tsallis-INF [Zimmert and Seldin, 2021], which is FTRL with 1/2-Tsallis entropy and $\eta_t \propto 1/\sqrt{t}$ learning rate.

### 5.1 TS-Prod

Recall the 1/2-Tsallis regularizer is $F(x) = -\sum_{i=1}^{K} 2\sqrt{x_i}$. Unlike OMD, FTRL is not canonically expressed as a 1-step update of the previous policy. Instead, the 1/2-Tsallis-INF policy is given with a normalization constant $\Lambda_t$ (see [Zimmert and Seldin, 2021])

$$\pi_{t+1,i}^{\text{FTRL}} = \left( \Lambda_{t+1} + \eta_t \sum_{s=1}^{t} \hat{\ell}_{s,i} \right)^{-2} \,.$$

Recursively using this expression yields

$$\pi_{t+1,i}^{\text{FTRL}} = \left(\Lambda_{t+1} + \frac{\eta_t}{\eta_{t-1}}\left(\frac{1}{\sqrt{\pi_{t,i}^{\text{FTRL}}}} - \Lambda_t\right) + \eta_t \hat{\ell}_t\right)^{-2} = \pi_{t,i}^{\text{FTRL}}\left(1 + \eta_t\sqrt{\pi_{t,i}^{\text{FTRL}}}\left(\hat{\ell}_t - \frac{\eta_t\xi_t}{\sqrt{\pi_{t,i}^{\text{FTRL}}}} - \lambda_t\right)\right)^{-2},$$

where $\xi_t = \frac{1}{\eta_t^2} - \frac{1}{\eta_t\eta_{t-1}}$ and $\lambda_t = \Lambda_{t+1} - \frac{\eta_t}{\eta_{t-1}}\Lambda_t$. The first order approximation is

$$\pi_{t+1,i}^{\text{FTRL}} \underbrace{\approx}_{\text{1st-order}} \pi_{t,i}^{\text{FTRL}}(1 - 2\eta_t\sqrt{\pi_{t,i}^{\text{FTRL}}}(\hat{\ell}_{t,i} - \xi_t/\sqrt{\pi_{t,i}^{\text{FTRL}}} - \lambda_t)).$$

We ensure following the approximation in expectation by directly biasing the losses with $\eta_t\xi_t/\sqrt{\pi_{t,i}}$. Additionally, one needs to perform a second-order correction as discussed in Section 3.1. We omit a formal derivation, but notice that WSU-UX required $\eta/\pi_{t,i}$ correction, while LB-Prod works without correction because the error is of order $\eta$. As the intermediate potential between KL and Logbarrier, Tsallis-INF turns out to require a correction of order $\eta/\sqrt{\pi_{t,i}}$, which we tighten by an additional factor of $(1 - \pi_{t,i})$ necessary to ensure stochastic bounds.

With this, we are ready to present

$$\hat{\ell}_{t,i} = \left(\ell_{t,i} - \frac{\eta_t(\xi_t + \gamma(1 - \pi_{t,i}))}{\sqrt{\pi_{t,i}}}\right)\frac{\mathbb{I}(A_t = i)}{\pi_{t,i}}, \quad \lambda_t = \sum_{i=1}^{K}\frac{\pi_{t,i}^{\frac{3}{2}}\hat{\ell}_{t,i}}{\sum_{j=1}^{K}\pi_{t,j}^{\frac{3}{2}}}, \quad \xi_t = \frac{1}{\eta_t^2} - \frac{1}{\eta_t\eta_{t-1}},$$

$$\pi_{t+1,i} = \pi_{t,i}(1 - 2\eta_t\sqrt{\pi_{t,i}}(\hat{\ell}_{t,i} - \lambda_t)). \tag{TS-Prod}$$

**Theorem 3.** *The regret of TS-Prod with $\eta_t = \frac{1}{\sqrt{K + 26t}}$, $\gamma = \frac{13}{2}$ is bounded by $O(\sqrt{KT} + K\log(T))$ in the adversarial setting and by $O\left(\sum_{i \neq i^\star}\frac{\log(T)}{\Delta_i}\right)$ in the stochastic setting.*

## 5.2   Analysis of TS-Prod

We first show that the loss biasing is sufficient to ensure that the distribution is well defined.

**Lemma 5.** *If $\xi_t$ is a non-increasing sequence, $\eta_t < \frac{2}{\sqrt{K(\xi_t + \gamma)^2}}$ and $\eta_{t+1}^2 \leq \eta_t^2(1 - 2\gamma\eta_t^2)$ for all $t$, then the update rule of TS-Prod is proper and satisfies $\pi_{t,i} > (\xi_t + \gamma)^2\eta_t^2$ for any arm and loss sequence at all time steps.*

Next we present the moving parts of the analysis.

$$\mathbb{E}[\sum_{t=1}^{T}\langle\pi_t - u, \ell_t\rangle] = \sum_{t=1}^{T}\mathbb{E}\left[\underbrace{\left\langle\pi_t - u, \ell_t - \hat{\ell}_t\right\rangle}_{\text{change of measure}} + \underbrace{\frac{D_{TS}(u, \pi_t) - D_{TS}(u, \pi_{t+1})}{\eta_t}}_{\text{proto-penalty}}\right.$$
$$\left.+ \underbrace{\left\langle\pi_t - u, \hat{\ell}_t\right\rangle - \frac{D_{TS}(u, \pi_t) - D_{TS}(u, \pi_{t+1})}{\eta_t}}_{\text{proto-stability}}\right]$$

The change of measure is by construction

$$\text{change-of-measure} = \sum_{t=1}^{T}\mathbb{E}\left[\sum_{i=1}^{K}\frac{\pi_{t,i} - u_i}{\sqrt{\pi_{t,i}}}\left(\eta_t\xi_t + \eta_t\gamma(1 - \pi_{t,i})\right)\right]. \tag{3}$$

We now bound the stability and penalty.

**Lemma 6.** *For any time $t$ such that $\pi_{t,i} > (\xi_t + \gamma)^2\eta_t^2$, it holds*

$$\left\langle\pi_t - u, \mathbb{E}_t[\hat{\ell}_t]\right\rangle + \mathbb{E}_t\left[\frac{D_{TS}(u, \pi_{t+1})}{\eta_t}\right] - \frac{D_{TS}(u, \pi_t)}{\eta_t} \leq \frac{13}{2}\frac{\eta_t u_i}{\sqrt{\pi_{ti}}}(1 - \pi_{ti}).$$

*The tuning of Theorem 3 satisfies the conditions.*

**Lemma 7.** *The proto-penalty is bounded by*

$$\sum_{t=1}^{T} \frac{D_{TS}(u, \pi_t) - D_{TS}(u, \pi_{t+1})}{\eta_t} \leq \sum_{t=1}^{T} \eta_t \xi_t \left( \sum_{i \neq i^\star} 2\sqrt{\pi_{ti}} + \sum_{i=1}^{K} \frac{u_i - \pi_{ti}}{\sqrt{\pi_{ti}}} \right)$$

We are ready to prove the main regret guarantee.

*Proof of Theorem 3.* By Lemma 5 we have a proper update rule. Using Lemma 6, equation (3), where we tuned $\gamma = \frac{13}{2}$ and Lemma 7 yields

$$\mathbb{E}\left[ \sum_{t=1}^{T} \langle \pi_t - u, \ell_t \rangle \right] \leq \mathbb{E}\left[ \sum_{t=1}^{T} \eta_t \left( \sum_{i=1}^{K} \frac{13}{2} \sqrt{\pi_{t,i}}(1 - \pi_{ti}) + \sum_{i \neq i^\star} \xi_t \sqrt{\pi_{t,i}} \right) \right].$$

The adversarial regret follows from $\sum_{i=1}^{K} \sqrt{\pi_{t,i}} \leq \sqrt{K}$, $\xi_t \leq 4$, $\sum_{t=1}^{T} \eta_t = O(\sqrt{T})$. For the stochastic regret, the proof follows standard arguments using the self-bounding trick as in Zimmert and Seldin [2021]. For details on the self-bounding trick see the supplementary material. □

### 5.3 TS-Prod and stabilized OMD

Even though we derived TS-Prod from FTRL, it turns out that one can also interpret the update as an approximation of *stabilized* OMD proposed by Fang et al. [2022]. In Appendix E, we formalize this connection and present a slight variation of TS-Prod. We then analyse this variant via the perturbation technique described in Section 4.3. Our analysis also shows that the stabilized OMD algorithm induced by the $1/2$-Tsallis entropy enjoys best-of-both worlds regret guarantees which to the best of our knowledge is novel.

## 6 Discussion

We have provided an extensive study of incentive-compatible bandits. We have negatively resolved an open question of whether incentive-compatibility as defined in Freeman et al. [2020] is harder than regular bandits. Using linear approximations, partly with second order corrections, allows to recover results from well studied algorithms in the literature. We even obtain an algorithm with best-of-both-world guarantees. Our algorithms are conceptually simpler than existing bandit algorithms, they update the probability distributions with basic arithmetic operations without the need to solve optimization problems.

Our successes make it likely that one can transfer even more sophisticated methods, such as first-order, second-order, path-norm bounds and online learning with graph feedback to this framework. We leave this investigation to future work.

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

## Contents

# A  Background on FTRL and OMD

We give a brief overview of the template analysis for FTRL and OMD. For an extensive discussion and regret analysis of these two frameworks we refer the interested readers to Chapter 2 of Shalev-Shwartz [2012], Chapter 5 of Hazan et al. [2016] or Chapter 28 of Lattimore and Szepesvári [2020].

## A.1  OMD analysis overview

The OMD update

$$\pi_{t+1} = \arg\min_{\pi \in \Delta([K])} \left\langle \pi, \eta_t \hat{\ell}_t \right\rangle - D_F(\pi, \pi_t) \tag{OMD}$$

can be written in two steps as

$$\tilde{\pi}_{t+1} = \arg\min_{\pi \in \mathbb{R}^K} \left\langle \pi, \eta_t \hat{\ell}_t \right\rangle - D_F(\pi, \pi_t)$$

$$\pi_{t+1} = \arg\min_{\pi \in \Delta([K])} D_F(\pi, \tilde{\pi}_{t+1}),$$

where the first step is an unconstrained optimization over the linear loss at time $t$ together with a regularization term given by the Bregman divergence induced by $F$, and the second step is the Bregman projection onto the probability simplex. Thie first step of the OMD update can be re-written as

$$\nabla F(\tilde{\pi}_{t+1}) = \nabla F(\pi_t) - \eta_t \hat{\ell}_t,$$

Let $F_t = \frac{1}{\eta_t}$. Since $\pi_{t+1}$ is the minimizer of the OMD update, and $D_F w$ is convex we have that

$$\left\langle \pi_{t+1} - \pi, \hat{\ell}_t \right\rangle \leq \langle u - \pi_{t+1}, \nabla F_t(\pi_{t+1}) - \nabla F_t(\pi_t) \rangle$$
$$= D_{F_t}(u, \pi_t) - D_{F_t}(u, \pi_{t+1}) - D_{F_t}(\pi_{t+1}, \tilde{\pi}_t).$$

Further, it holds that

$$\left\langle \pi_t - \pi_{t+1}, \hat{\ell}_t \right\rangle = D_{F_t}(\pi_{t+1}, \pi_t) + D_{F_t}(\pi_t, \tilde{\pi}_{t+1}) - D_{F_t}(\tilde{\pi}_{t+1}, \pi_{t+1})$$
$$\leq D_{F_t}(\pi_{t+1}, \pi_t) + D_{F_t}(\pi_t, \tilde{\pi}_{t+1}).$$

Combining the two inequalities together we have that one step of the regret to any $u \in \Delta([K])$ is bounded as

$$\left\langle \hat{\ell}_t, \pi_t - u \right\rangle \leq D_{F_t}(u, \pi_t) - D_{F_t}(u, \pi_{t+1}) + D_{F_t}(\pi_t, \tilde{\pi}_{t+1}).$$

For a fixed step-size $\eta_t = \eta$ the above telescopes to bound the regret as

$$\sum_{t=1}^{T} \left\langle \hat{\ell}_t, \pi_t - u \right\rangle \leq \frac{D_F(u, \pi_1) - D_F(u, \pi_T)}{\eta} + \frac{1}{\eta} \sum_{t=1}^{T-1} D_F(\pi_t, \tilde{\pi}_{t+1}).$$

As long as $F$ is twice differentiable, each of the terms can be bounded as $D_F(\pi_t, \tilde{\pi}_{t+1}) \leq O(\eta^2 \|\hat{\ell}_t\|^2_{\nabla^2(F^*)(w_t)})$, where $F^*$ is the Fenchel conjugate of $F$. Controlling $\|\hat{\ell}_t\|^2_{\nabla^2(F^*)(w_t)}$ in OCO is usually done by assuming some boundedness of the losses. In bandit literature controlling this term is slightly more involved and depends on the choice of $F$.

When $\eta$ is not constant, telescoping the above sum does not work and the analysis becomes much more involved. It is possible to construct sequences of losses for which the OMD update does not enjoy sub-linear regret for $\eta_t = \frac{1}{\sqrt{t}}$. Fang et al. [2022] introduce a stabilization term to the OMD update which overcomes this problem and show that this new version does enjoy the standard OMD regret guarantees.

## A.2  FTRL analysis overview

The FTRL analysis follows similar ideas, however, the one step regret is bounded as

$$\left\langle \hat{\ell}_t, \pi_t - u \right\rangle \leq (F_t + I_{\Delta([K])})^*(-\hat{L}_{t-1}) - (F_t + I_{\Delta([K])})^*(-\hat{L}_t) + \frac{1}{\eta_t} D_{F^*}(-\hat{L}_t, -\hat{L}_{t-1}),$$

where $\hat{L}_t = \sum_{s=1}^{t-1} \hat{\ell}_t$, and $I_{\Delta[K]}$ is the characteristic function of $\Delta_{[K]}$, i.e., $I_{\Delta[K]}(\pi) = 0$ if $\pi \in \Delta[K]$ and $I_{\Delta[K]}(\pi) = +\infty$ otherwise. The term $D_{F^*}(-\hat{L}_t, -\hat{L}_{t-1})$ can be thought of as the equivalent to $D_F(\pi_t, \tilde{\pi}_{t+1})$ in the OMD analysis. The term $(F_t + I_{\Delta([K])})^*(-\hat{L}_{t-1}) - (F_t + I_{\Delta([K])})^*(-\hat{L}_t)$ needs to be telescoped in an appropriate way. For more details we refer the reader to the penalty term bound of Zimmert and Seldin [2021].

## B  Missing Proof Section 3

*Proof of Theorem 1.* WLOG we assume that $T \geq K$. We begin with the bound from Lemma 2. The second and third term in the RHS of the inequality are bounded as is standard in the Exp3 analysis

$$\eta \sum_{t=1}^{T} \sum_{i=1}^{K} \pi_{t,i} \, \mathbb{E}[\hat{\ell}_{t,i}^2 | \mathcal{F}_{t-1}] \leq \frac{\eta T K}{1 - \gamma} \leq 2\eta T K, \qquad \eta \sum_{t=1}^{T} \mathbb{E}[\hat{\ell}_{t,i^*}^2] \leq \eta \sum_{t=1}^{T} \mathbb{E}\left[\frac{\tilde{\ell}_{t,i^*}^2}{\tilde{\pi}_{t,i^*}}\right] \leq \eta \sum_{t=1}^{T} \mathbb{E}\left[\frac{\ell_{t,i^*}^2}{\tilde{\pi}_{t,i^*}}\right],$$

where $\mathcal{F}_{t-1}$ is the filtration generated by the random play and randomness of the losses up to time $t-1$. We now consider the expectation of the LHS which evaluates to

$$\sum_{t=1}^{T} \sum_{i=1}^{K} \mathbb{E}[\pi_{t,i} \hat{\ell}_{t,i}] - \sum_{t=1}^{T} \mathbb{E}[\hat{\ell}_{t,i^*}] = \sum_{t=1}^{T} \sum_{i=1}^{K} \mathbb{E}[\pi_{t,i} \tilde{\ell}_{t,i}] - \sum_{t=1}^{T} \mathbb{E}[\tilde{\ell}_{t,i^*}] = \sum_{t=1}^{T} \sum_{i=1}^{K} \mathbb{E}[\pi_{t,i} \ell_{t,i}] - \sum_{t=1}^{T} \mathbb{E}[\ell_{t,i^*}]$$

$$+ \sum_{t=1}^{T} \mathbb{E}\left[\frac{\eta \ell_{t,i^*}}{\tilde{\pi}_{t,i}}\right] - \sum_{t=1}^{T} \sum_{i=1}^{K} \mathbb{E}\left[\frac{\eta \pi_{t,i} \ell_{t,i}}{\tilde{\pi}_{t,i}}\right] \geq \eta \sum_{t=1}^{T} \mathbb{E}\left[\frac{\ell_{t,i^*}^2}{\tilde{\pi}_{t,i}}\right] - 2\eta T K$$

$$+ \sum_{t=1}^{T} \sum_{i=1}^{K} \mathbb{E}[\pi_{t,i} \ell_{t,i}] - \sum_{t=1}^{T} \mathbb{E}[\ell_{t,i^*}].$$

Thus combining the bounds on the LHS and RHS we have

$$\sum_{t=1}^{T} \sum_{i=1}^{K} \mathbb{E}[\pi_{t,i} \ell_{t,i}] - \sum_{t=1}^{T} \mathbb{E}[\ell_{t,i^*}] \leq \frac{\log(K)}{\eta} + 4\eta T K + \eta \sum_{t=1}^{T} \mathbb{E}\left[\frac{\ell_{t,i^*}^2}{\tilde{\pi}_{t,i}}\right] - \eta \sum_{t=1}^{T} \mathbb{E}\left[\frac{\ell_{t,i^*}^2}{\tilde{\pi}_{t,i}}\right].$$

To complete the proof we only note that $\sum_{t=1}^{T} \sum_{i=1}^{K} \mathbb{E}[\pi_{t,i} \ell_{t,i}] - \sum_{t=1}^{T} \mathbb{E}[\ell_{t,A_t}] \leq 2T\gamma = \eta K T$. $\qquad\square$

## C  Missing Proofs Section 4

*Proof of Lemma 3.* The expectation of $\tilde{\ell}_{ti} = \ell_{ti} \mathbb{I}(A_t = i)$ is obviously $\pi_{ti} \ell_{ti}$, hence by the definition of $\lambda_{ti}$, we have

$$\mathbb{E}_t[\tilde{\ell}_{ti} - \lambda_{ti}] = \pi_{ti}\left(\ell_{ti} - \frac{\sum_{j=1}^{K} \pi_{tj}^2 \ell_{tj}}{\sum_{j=1}^{K} \pi_{tj}^2}\right).$$

For the second part, we have

$$\mathbb{E}_t[(\tilde{\ell}_{ti} - \lambda_{ti})^2] \leq \mathbb{E}_t[\tilde{\ell}_{ti}^2] + \mathbb{E}_t[\lambda_{ti}^2] = \pi_{ti}\left(\ell_{ti}^2 + \pi_{ti}^2 \frac{\sum_{j=1}^{K} \pi_{tj}^3 \ell_{tj}^2}{\left(\sum_{k=1}^{K} \pi_{tk}^2\right)^2}\right) \leq \pi_{ti}\left(1 + \frac{\pi_{ti} \sum_{j=1}^{K} \pi_{tj}^3}{\left(\sum_{k=1}^{K} \pi_{tk}^2\right)^2}\right).$$

The proof is completed by noting

$$\frac{\pi_{ti} \sum_{j=1}^{K} \pi_{tj}^3}{\left(\sum_{k=1}^{K} \pi_{tk}^2\right)^2} \leq \frac{\pi_{ti} \sum_{j=1}^{K} \pi_{tj}^3}{\left(\sum_{k=1}^{K} \pi_{tk}^3\right)^{\frac{4}{3}}} = \frac{\pi_{ti}}{\left(\sum_{k=1}^{K} \pi_{tk}^3\right)^{\frac{1}{3}}} \leq 1.$$

$\qquad\square$

*Proof of Lemma 4.*

$$\langle \pi_t - u, \ell_t \rangle + \mathbb{E}_t \left[ \eta^{-1} D_{LB}(u, \pi_{t+1}) \right] - \eta^{-1} D_{LB}(u, \pi_t)$$

$$= \langle \pi_t - u, \ell_t \rangle + \mathbb{E}_t \left[ \sum_{i=1}^{K} \frac{u_i - \pi_{t+1,i}}{\eta \pi_{t+1,i}} - \frac{u_i - \pi_{t,i}}{\eta \pi_{t,i}} + \frac{1}{\eta} \log \left( \frac{\pi_{t+1,i}}{\pi_{t,i}} \right) \right]$$

$$= \langle \pi_t - u, \ell_t \rangle + \mathbb{E}_t \left[ \sum_{i=1}^{K} \frac{u_i \left( 1 - \frac{\pi_{t+1,i}}{\pi_{t,i}} \right)}{\eta \pi_{t+1,i}} + \frac{1}{\eta} \log \left( 1 - \eta(\tilde{\ell}_{t,i} - \lambda_{t,i}) \right) \right]$$

$$\leq \langle \pi_t - u, \ell_t \rangle + \mathbb{E}_t \left[ \sum_{i=1}^{K} \frac{u_i(\tilde{\ell}_{t,i} - \lambda_{t,i})}{\pi_{t,i}(1 - \eta(\tilde{\ell}_{t,i} - \lambda_{t,i}))} - \tilde{\ell}_{t,i} + \lambda_{t,i} \right] \qquad (\log(1+x) \leq x)$$

$$\leq \langle \pi_t - u, \ell_t \rangle + \sum_{i=1}^{K} \left( \frac{u_i}{\pi_{t,i}} - 1 \right) \mathbb{E}_t[\tilde{\ell}_{t,i} - \lambda_{t,i}] + \eta \sum_{i=1}^{K} \frac{u_i}{\pi_{t,i}} \mathbb{E}_t \left[ \frac{(\tilde{\ell}_{t,i} - \lambda_{t,i})^2}{1 - \eta} \right]$$

$$\leq \langle \pi_t - u, \ell_t \rangle + \sum_{i=1}^{K} (u_i - \pi_{t,i})(\ell_{t,i} - c_t) + \frac{2\eta}{1 - \eta} \sum_{i=1}^{K} u_i = \frac{2\eta}{1 - \eta}. \qquad (\text{Lemma 3})$$

$\square$

# D    Missing Proofs Section 5.2

## D.1    Technical Lemmas

**Lemma 8.**

$$\min_{x \in [0,1]} f(x) = \min_{x \in [0,1]} \frac{x^3}{1-x} + \sqrt{1-x} \geq \sqrt{\frac{8}{9}}.$$

*Proof.* We first show that the optimal point is smaller than $\frac{1}{9}$, by looking at the derivative

$$f'(x) = \frac{3\sqrt{x} - x^{\frac{3}{2}} - (1-x)^{\frac{3}{2}}}{2(1-x)^2}.$$

For the enumerator, we have for all $x \geq \frac{1}{9}$:

$$3\sqrt{x} - x^{\frac{3}{2}} - (1-x)^{\frac{3}{2}} > 3\sqrt{x} - \max\{\sqrt{x}, \sqrt{1-x}\} \qquad \geq \min\{2\sqrt{x}, 3\sqrt{x} - 1\} \geq 0$$

Hence

$$\min_{x \in [0,1]} f(x) = \min_{x \in [0,1/9]} f(x) > \min_{x \in [0,1/9]} \sqrt{1-x} = \sqrt{\frac{8}{9}}.$$

$\square$

**Lemma 9.** *For any $a, b \geq 0$ such that $a + b \geq 1$, it holds*

$$\frac{a}{\sqrt{b}} + \frac{b}{\sqrt{a}} \geq \sqrt{2}.$$

*Proof.* We can assume w.l.o.g. that $a = 1 - b$, otherwise scale both $a$ and $b$ down and reduce the objective. The resulting problem is symmetric with $a = \frac{1}{2}$ as the unique minimizer resulting in the statement. $\square$

## D.2 Minimal probability

**Lemma 10.** *Assume that $\pi_{ti} > (\xi_t + \gamma)^2 \eta_t^2$ holds for all arms, then*

$$\mathbb{E}_t\left[(\hat{\ell}_{ti} - \lambda_t)^2\right] \le \frac{13(1 - \pi_{ti})}{8\pi_{t,i}}\,.$$

*Proof.* Let $\tilde{\ell}_{t,i} = \pi_{t,i}\hat{\ell}_{t,i}$ and $\lambda_{t,i} = \pi_{t,i}\lambda_t$. Note that $\tilde{\ell}_{ti} \in [-1, 1]$ by the condition on $\pi_{ti}$ and $\tilde{\ell}_{ti} = 0$ for $i \ne I_t$ by construction of the loss estimate. Hence

$$
\mathbb{E}_t\left[(\tilde{\ell}_{ti} - \lambda_{ti})^2\right] \le \pi_{ti}\left(1 - \frac{\pi_{ti}^{\frac{3}{2}}}{\sum_{j=1}^K \pi_{tj}^{\frac{3}{2}}}\right)^2 \mathbb{E}[\tilde{\ell}_{ti}^2 | i = I_t] + \sum_{j \ne i} \pi_{tj}\left(\frac{\pi_{ti}\sqrt{\pi_{tj}}}{\sum_{j=1}^K \pi_{tj}^{\frac{3}{2}}}\right)^2 \mathbb{E}[\tilde{\ell}_{tj}^2 | j = I_t]
$$

$$
\le \pi_{ti}\left(\frac{(\sum_{j \ne i} \pi_{tj}^{\frac{3}{2}})^2 + \pi_{ti}\sum_{j \ne i} \pi_{tj}^2}{(\sum_{j=1}^K \pi_{tj}^{\frac{3}{2}})^2}\right)
$$

$$
= \pi_{ti}(1 - \pi_{ti})\left(\frac{(1 - \pi_{ti})^2(\sum_{j \ne i} \tilde{\pi}_{tj}^{\frac{3}{2}})^2 + \pi_{ti}(1 - \pi_{ti})\sum_{j \ne i} \tilde{\pi}_{tj}^2}{(\pi_{ti}^{\frac{3}{2}} + (1 - \pi_{ti})^{\frac{3}{2}}\sum_{j \ne i} \tilde{\pi}_{tj}^{\frac{3}{2}})^2}\right),
$$

where $\tilde{\pi}_{tj} = \frac{\pi_{tj}}{1 - \pi_{ti}}$. We bound the two terms in the bracket separately, for the first term we have

$$
\left(\frac{(1 - \pi_{ti})\sum_{j \ne i} \tilde{\pi}_{tj}^{\frac{3}{2}}}{\pi_{ti}^{\frac{3}{2}} + (1 - \pi_{ti})^{\frac{3}{2}}\sum_{j \ne i} \tilde{\pi}_{tj}^{\frac{3}{2}}}\right)^2 \le \left(\frac{(1 - \pi_{ti})}{\pi_{ti}^{\frac{3}{2}} + (1 - \pi_{ti})^{\frac{3}{2}}}\right)^2 \qquad (\sum_{j \ne i} \tilde{\pi}_{tj} = 1)
$$

$$
\le \frac{9}{8} \qquad \text{(Lemma 8)}
$$

The second term is

$$
\frac{\pi_{ti}(1 - \pi_{ti})\sum_{j \ne i} \tilde{\pi}_{tj}^2}{(\pi_{ti}^{\frac{3}{2}} + (1 - \pi_{ti})^{\frac{3}{2}}\sum_{j \ne i} \tilde{\pi}_{tj}^{\frac{3}{2}})^2} \le \frac{\pi_{ti}(1 - \pi_{ti})(\sum_{j \ne i} \tilde{\pi}_{tj}^{\frac{3}{2}})^{\frac{4}{3}}}{(\pi_{ti}^{\frac{3}{2}} + (1 - \pi_{ti})^{\frac{3}{2}}\sum_{j \ne i} \tilde{\pi}_{tj}^{\frac{3}{2}})^2}
$$

$$
= \left(\frac{\pi_{ti}}{\sqrt{1 - \pi_{ti}}(\sum_{j \ne i} \tilde{\pi}_{tj}^{\frac{3}{2}})^{\frac{2}{3}}} + \frac{(1 - \pi_{ti})(\sum_{j \ne i} \tilde{\pi}_{tj}^{\frac{3}{2}})^{\frac{1}{3}}}{\sqrt{\pi_{ti}}}\right)^{-2}
$$

$$
\le \frac{1}{2} \qquad \text{(Lemma 9)}
$$

$\square$

**Lemma 11** (Lemma 5). *If $\xi_t$ is a non-increasing sequence, $\eta_t < \frac{2}{\sqrt{K(\xi_t + \gamma)^2}}$ and $\eta_{t+1}^2 \le \eta_t^2(1 - 2\gamma\eta_t^2)$ for all t, then the update rule of TS-Prod is well defined and satisfies $\pi_{t,i} > (\xi_t + \gamma)^2 \eta_t^2$ for any arm and loss sequence at all time steps.*

*Proof.* The proof follows by induction. At $t = 1$ the statement is true by definition. Let the claim hold at time $t$, then the probability of an arm only decreases when $\hat{\ell}_{t,i} - \lambda_t$ is positive. We look at the cases where $A_t = i$ and $A_t \ne i$ independently.

**Case $A_t = i$:**

$$
\pi_{t+1,i} > \pi_{t,i}(1 - 2\eta_t\sqrt{\pi_{t,i}}\hat{\ell}_{t,i}) = \pi_{t,i} - 2\eta_t\left(\sqrt{\pi_{t,i}}\ell_{t,i} - \eta_t(\xi_t + \gamma(1 - \pi_{t,i}))\right)
$$

$$
> \pi_{t,i}(1 - 2\gamma\eta_t^2) - 2\eta_t\sqrt{\pi_{t,i}} + 2\gamma\eta_t^2
$$

$$
= (1 - 2\gamma\eta_t^2)\left(\sqrt{\pi_{t,i}} - \frac{\eta_t}{1 - 2\gamma\eta_t^2}\right)^2 + (2\gamma - \frac{1}{1 - 2\gamma\eta_t^2})\eta_t^2
$$

This is a quadratic function in $\pi_{t,i}$ with minimizer $\frac{\eta_t^2}{(1-2\gamma\eta_t^2)^2} < (\xi_t + \gamma)^2\eta_t^2$, hence the value is lower bounded by setting $\pi_{t,i}$ to $(\xi_t + \gamma)^2\eta_t^2$

$$\pi_{t+1,i} > (\xi_t + \gamma)^2\eta_t^2(1 - 2\gamma\eta_t^2) \geq (\xi_t + \gamma)^2\eta_{t+1}^2 \geq (\xi_{t+1} + \gamma)^2\eta_{t+1}^2$$

**Case $A_t \neq i$:**

$$\pi_{t+1,i} = \pi_{t,i} - 2\eta_t \pi_{t,i}^{\frac{3}{2}} \left( \frac{\eta_t((\xi_t + \gamma(1 - \pi_{t,A_t})) - \ell_{t,A_t}\sqrt{\pi_{t,A_t}}}{\sum_{j=1}^{K} \pi_{tj}^{\frac{3}{2}}} \right)$$

$$> \pi_{t,i} - 2(\xi_t + \gamma)\eta_t^2 \pi_{t,i}^{\frac{3}{2}}\sqrt{K}$$

This is a concave function (in $\pi_{t,i}$) so the minimizer is at either at $\pi_{t,i} = (\xi_t + \gamma)^2\eta_t^2$ or at $\pi_{t,i} = 1$. For the latter, we have have $\pi_{t+1,i} > \frac{1}{2}$, so the minimum is obtained for the first case.

$$\pi_{t+1,i} > (\xi_t + \gamma)^2\eta_t^2 - 2(\xi_t + \gamma)^4\eta_t^5\sqrt{K} > (\xi_t + \gamma)^2\eta_t^2(1 - 2\gamma\eta_t^2) \geq C_{t+1}^2\eta_{t+1}^2 \,.$$

$\square$

**Lemma 12** (Lemma 6). *For any time $t$ such that $\pi_{t,i} > (\xi_t + \gamma)^2\eta_t^2$, it holds*

$$\langle\pi_t - u, \mathbb{E}_t[\ell_t]\rangle + \mathbb{E}_t\left[\frac{D_{TS}(u, \pi_{t+1})}{\eta_t}\right] - \frac{D_{TS}(u, \pi_t)}{\eta_t} \leq \sum_{i=1}^{K}\left(2\eta_t\sqrt{\pi_{t,i}}(1 - \pi_{t,i}) - \left(\frac{1}{\eta_t} - \frac{1}{\eta_{t-1}}\right)\frac{u_i - \pi_{t,i}}{\sqrt{\pi_{t,i}}}\right) \,.$$

*Proof of Lemma 6.*

$$\langle\pi_t - u, \mathbb{E}_t[\ell_t]\rangle + \mathbb{E}_t\left[\eta_t^{-1}D_{TS}(u, \pi_{t+1})\right] - \eta_t^{-1}D_{TS}(u, \pi_t)$$

$$= \langle\pi_t - u, \mathbb{E}_t[\ell_t]\rangle + \mathbb{E}_t\left[\sum_{i=1}^{K}\frac{u_i - \pi_{t+1,i}}{\eta_t\sqrt{\pi_{t+1,i}}} - \frac{u_i - \pi_{ti}}{\eta_t\sqrt{\pi_{ti}}} + \frac{1}{\eta_t}\left(2\sqrt{\pi_{t+1,i}} - 2\sqrt{\pi_{ti}}\right)\right]$$

$$= \langle\pi_t - u, \mathbb{E}_t[\ell_t]\rangle + \sum_{i=1}^{K}\left(\frac{u_i}{\eta_t\sqrt{\pi_{ti}}}\mathbb{E}_t\left[\sqrt{\frac{\pi_{ti}}{\pi_{t+1,i}}} - 1\right] + \frac{\pi_{ti}}{\eta_t\sqrt{\pi_{ti}}}\mathbb{E}_t\left[\sqrt{\frac{\pi_{t+1,i}}{\pi_{ti}}} - 1\right]\right)$$

$$= \langle\pi_t - u, \mathbb{E}_t[\ell_t]\rangle + \sum_{i=1}^{K}\left(\frac{u_i}{\eta_t\sqrt{\pi_{ti}}}\mathbb{E}_t\left[\sqrt{1 + \frac{2\eta_t\sqrt{\pi_{ti}}(\hat{\ell}_{ti} - \lambda_t)}{1 - 2\eta_t\sqrt{\pi_{ti}}(\hat{\ell}_{ti} - \lambda_t)}} - 1\right]\right.$$

$$\left. + \frac{\pi_{ti}}{\eta_t\sqrt{\pi_{ti}}}\mathbb{E}_t\left[\sqrt{1 - 2\eta_t\sqrt{\pi_{ti}}(\hat{\ell}_{ti} - \lambda_t)} - 1\right]\right)$$

$$\leq \langle\pi_t - u, \mathbb{E}_t[\ell_t]\rangle + \sum_{i=1}^{K}\left(\frac{u_i}{\eta_t\sqrt{\pi_{ti}}}\mathbb{E}_t\left[\eta_t\sqrt{\pi_{ti}}(\hat{\ell}_{ti} - \lambda_t) + \frac{2\eta_t^2\pi_{ti}(\hat{\ell}_{ti} - \lambda_t)^2}{1 - 2\eta_t\sqrt{\pi_{ti}}(\hat{\ell}_{ti} - \lambda_t)}\right]\right.$$

$$\left. + \frac{\pi_{ti}}{\eta_t\sqrt{\pi_{ti}}}\mathbb{E}_t\left[-\eta_t\sqrt{\pi_{ti}}(\hat{\ell}_{ti} - \lambda_t)\right]\right)$$

$$\leq \langle\pi_t - u, \mathbb{E}_t[\ell_t]\rangle + \sum_{i=1}^{K}\left((u_i - \pi_{ti})\mathbb{E}_t\left[\hat{\ell}_{ti} - \lambda_t\right] + 4\eta_t u_{ti}\sqrt{\pi_{ti}}\mathbb{E}_t\left[(\hat{\ell}_{ti} - \lambda_t)^2\right]\right)$$

$$\text{(Setting } (\xi_t + \gamma) \geq 4)$$

$$\leq \frac{13}{2}\frac{\eta_t u_i}{\sqrt{\pi_{ti}}}(1 - \pi_{ti}) \,. \qquad \text{(Lemma 10)}$$

We now show that the requirements of Lemma 5 are satisfied with the tuning of Theorem 3. With $\eta_t = \frac{1}{\sqrt{K+26t}}$, we have $c_t = \left(K + 26t - \sqrt{(K + 26t)(K + 26t - 26)}\right) > 2$, which is monotonically decreasing. $c_t > 2$ and $\gamma = \frac{13}{2}$ ensures that $\eta_t = \frac{1}{\sqrt{K+26t}} < \frac{2}{\sqrt{K(c_t+\gamma)^2}}$. Further we have

$$\frac{\eta_{t+1}^2}{\eta_t^2} = \frac{K + 26t}{K + 26(t+1)} = 1 - \frac{26}{K + 26(t+1)} \leq 1 - \frac{4}{K + 26t} = 1 - 4\eta_t^2 \,.$$

$\square$

*Proof of Lemma 7.* Using $\eta_0 = \infty$ and $D_{TS}(u, \Pi_{T+1}) \geq 0$

$$\sum_{t=1}^{T} \frac{D_{TS}(u, \pi_t) - D_{TS}(u, \pi_{t+1})}{\eta_t} \leq \sum_{t=1}^{T} \left( \frac{1}{\eta_t} - \frac{1}{\eta_{t-1}} \right) D_{TS}(u, \pi_t)$$

$$= \sum_{t=1}^{T} \left( \frac{1}{\eta_t} - \frac{1}{\eta_{t-1}} \right) \left( F(u) - F(\pi_t) - \langle u - \pi_t, \nabla F(\pi_t) \rangle \right)$$

$$= \sum_{t=1}^{T} \sum_{i=1}^{K} \eta_t \xi_t \left( 2(\sqrt{\pi_{t,i}} - \sqrt{u_{t,i}}) - (u_i - \pi_{ti}) \frac{1}{\sqrt{\pi_{ti}}} \right)$$

$$\leq \sum_{t=1}^{T} \eta_t \xi_t \left( \sum_{i \neq i^\star} 2\sqrt{\pi_{t,i}} - \sum_{i=1}^{K} (u_i - \pi_{ti}) \frac{1}{\sqrt{\pi_{ti}}} \right),$$

where the last inequality follows from $\sum_{i=1}^{K} \sqrt{u_i} \geq 1 \geq \sqrt{\pi_{t,i^\star}}$. $\qquad\square$

### D.3 Self-bounding trick

We now quickly describe how to apply the self-bounding trick from Zimmert and Seldin [2021]. Assume that we have a regret bound of the form

$$\sum_{t=1}^{T} \sum_{i \neq i^*} \pi_{t,i} \Delta_i \leq \sum_{t=1}^{T} \sum_{i \neq i^*} \frac{a\pi_{t,i} + b\sqrt{\pi_{t,i}}}{\sqrt{t}},$$

for some positive $a$ and $b$. The above inequality implies

$$\frac{1}{3} \sum_{t=1}^{T} \sum_{i \neq i^*} \pi_{t,i} \Delta_i \leq \sum_{t=1}^{T} \sum_{i \neq i^*} \pi_{t,i} \left( \frac{a}{\sqrt{t}} - \frac{\Delta_i}{3} \right) + \sqrt{\pi_{t,i}} \left( \frac{b}{\sqrt{t}} - \frac{\Delta_i \sqrt{\pi_{t,i}}}{3} \right).$$

For a fixed $i$ the term $\left( \frac{a}{\sqrt{t}} - \frac{\Delta_i}{3} \right) \leq 0$ if $t \geq \frac{9a^2}{\Delta_i^2}$ and so the maximum regret from

$$\sum_{t=1}^{T} \pi_{t,i} \left( \frac{a}{\sqrt{t}} - \frac{\Delta_i}{3} \right) \leq \sum_{t=1}^{\lfloor \frac{9a^2}{\Delta_i^2} \rfloor} \frac{a}{\sqrt{t}} \leq \frac{6a^2}{\Delta_i}.$$

Further the term $\sqrt{\pi_{t,i}} \left( \frac{b}{\sqrt{t}} - \frac{\Delta_i \sqrt{\pi_{t,i}}}{3} \right) \leq \frac{2b^2}{t\Delta_i}$ for $t \geq \frac{4b^2}{\Delta_i^2}$. This implies

$$\sum_{t=1}^{T} \sqrt{\pi_{t,i}} \left( \frac{b}{\sqrt{t}} - \frac{\Delta_i \sqrt{\pi_{t,i}}}{3} \right) \leq \sum_{t=1}^{\lfloor \frac{4b^2}{\Delta_i^2} \rfloor} \frac{b}{\sqrt{t}} + \sum_{t=1}^{T} \frac{2b^2}{\Delta_i t} \leq \frac{8b^2}{\Delta_i} + \frac{2b^2 \log(T)}{\Delta_i}.$$

Combining the two bounds we have

$$\sum_{t=1}^{T} \sum_{i \neq i^*} \pi_{t,i} \Delta_i \leq O \left( \sum_{i \neq i^*} \frac{b^2 \log(T)}{\Delta_i} + \frac{a^2}{\Delta_i} \right).$$

## E  TS-Prod and stabilized OMD

The TS-Prod update is a linearization of the FTRL update as explained in Section 5.2. In this section we show a regret bound for the TS-Prod algorithm by linearizing the OMD update as we did for WSU-UX and LB-Prod. This comes with its own set of challenges. First, we believe that a decreasing step-size in the OMD update is important for achieving optimal regret bounds in the stochastic setting. Second, the vanilla OMD update with decreasing step-size might incur linear regret in the adversarial setting. This second issue is resolved by the *stabilized* OMD algorithms proposed by Fang et al. [2022]. TS-Prod turns out to be equivalent to the dual stabilized OMD algorithm of Fang et al. [2022]

and hence will inherit the adversarial regret guarantees. In the rest of the section we sketch the regret analysis for the stochastic setting and how we can reduce the regret analysis in the adversarial setting to that of Fang et al. [2022].

Stabilization as introduced by Fang et al. [2022] is the process of mixing the gradient mapping, $\nabla F(\pi_t)$, of the current iterate with the gradient mapping of the first iterate, $\nabla F(\pi_1)$, in the mirror descent update in the dual space, where $F(x) = -\sum_{i=1}^{K} 2\sqrt{x_i}$. This mixing turns out to be equivalent to the negative biasing of losses in Equation TS-Prod by the $\xi_t$ dependent terms.

The regret analysis begins by defining the perturbations $\{\epsilon_{t,i}\}_{t \in [T], i \in [K]}$ so that

$$\pi_{t+1,i} = \frac{\pi_{t,i}}{(1 + \eta_{t+1}\sqrt{\pi_{t,i}}(\hat{L}_{t,i} + \epsilon_{t,i}))^2}, \tag{4}$$

that is $\epsilon_{t,i}$ makes the update in Equation TS-Prod equivalent to the $1/2$-Tsallis mirror descent update. We note that the $\epsilon_{t,i}$ is only defined to assist with the regret analysis and it never needs to be computed for the actual update. The perturbations, $\epsilon_{t,i}$, are well controlled as we show next.

**Lemma 13.** *For every $t \in [T], i \in [K]$, there exists $\epsilon_{t,i}$ such that $\epsilon_{t,i} \leq 4\eta_t\sqrt{\pi_{t,i}}\hat{L}_{t,i}^2$ for all $i$ and $\ell_{t,i}$.*

Lemma 13 allows us to proceed with the analysis for the stochastic and adversarial cases by using the standard regret decomposition into a *penalty* and *stability* terms. In the stochastic case we can bound the two terms in the following way

**Lemma 14.** *For stochastic losses the penalty term is bounded in expectation by*

$$O\left(\frac{\mathbb{E}\left[\left(\sum_{i \neq i^*} \pi_{t+1,i}\right)^2\right]\sqrt{K}\log(t)}{\sqrt{t}} + \frac{1}{\sqrt{t}} \wedge \frac{\mathbb{E}\left[\left(\sum_{i \neq i^*} \pi_{t+1,i}\right)^2\right]\log(KT)}{\sqrt{t}}\right).$$

**Lemma 15.** *For stochastic losses the stability term is bounded by*

$$O\left(\frac{1}{\sqrt{t}}\sum_{i=1}^{K}\sqrt{\pi_{t,i}}(1 - \pi_{t,i}) + \frac{K\sqrt{\pi_{t,i}}}{t^2} + \frac{K}{t}\right).$$

The stochastic regret bound proof can now be completed by a careful self-bounding argument.

In the adversarial case we reduce the regret bound to that of Fang et al. [2022] in the following way. Let $\Phi = F + I_{\Delta^{K-1}}$ be potential defined by mixing the $1/2$-Tsallis potential together with the indicator function for the probability simplex. The update of Algorithm 2 (Dual Stabilized OMD) can then be written as

$$\begin{aligned}
\hat{w}_{t+1} &= \nabla\Phi(\pi_t) - \eta_t(\tilde{\ell}_t + \epsilon_t), \\
\hat{y}_{t+1} &= \chi_t\hat{w}_{t+1} + (1 - \chi_t)\nabla\Phi(\pi_1), \\
\pi_{t+1} &= \nabla\Phi^*(\hat{y}_{t+1}).
\end{aligned}$$

It turns out that this update is equivalent to the OMD update with respect to $\hat{L}_{t,i}$ in Equation 4. This allows us to use the regret bound in Theorem 3 [Fang et al., 2022]. Overall the regret of the perturbed OMD version is bounded as follows.

**Theorem 4.** *The regret of the algorithm defined by the update in Equation 4 is bounded by*

$$O\left(\sum_{i \neq i^*} \frac{\log(T)}{\Delta_i} + \frac{K\log^2(1/\Delta_{min})}{\Delta_{min}} + K^{3/2}\right)$$

*in the stochastic case, where $\Delta_{min}$ is the smallest gap between the expected losses. Further the regret in the adversarial setting is bounded by $O(\sqrt{KT})$.*

### E.1  Proof of Theorem 4

**Proof of Lemma 13.**  We work under the following assumption which is satisfied with the choice of $\eta_t$ and $\gamma$ by Lemma 5. Further, we are going to work with the following slight modification of the

losses $\hat{\ell}_t$ in the update of TS-Prod:

$$\hat{\ell}_{t,i} = \left(\ell_{t,i} - \frac{\eta_t(\gamma(1 - \pi_{t,i}))}{\sqrt{\pi_{t,i}}}\right) \frac{\mathbb{I}(A_t = i)}{\pi_{t,i}} - \frac{\eta_t \xi_t}{\sqrt{\pi_{t,i}}}.$$

We quickly check that the variance with this definition of $\hat{\ell}_t$ is bounded by Lemma 10

$$\mathbb{E}_t\left[\left(\left(\ell_{t,i} - \frac{\gamma(1 - \pi_{t,i})}{\sqrt{\pi_{t,i}}}\right) \frac{\mathbb{I}(A_t = i)}{\pi_{t,i}} - \frac{\eta_t \xi_t}{\sqrt{\pi_{t,i}}}\right)^2\right]$$

$$= \mathbb{E}_t\left[\left(\ell_{t,i} - \frac{\gamma(1 - \pi_{t,i})}{\sqrt{\pi_{t,i}}}\right)^2 \frac{1}{\pi_{t,i}} + \frac{(\eta_t \xi_t)^2}{\pi_{t,i}} - 2\left(\ell_{t,i} - \frac{\gamma(1 - \pi_{t,i})}{\sqrt{\pi_{t,i}}}\right) \frac{\eta_t \xi_t}{\sqrt{\pi_{t,i}}}\right]$$

$$\leq \mathbb{E}_t\left[\left(\ell_{t,i} - \frac{\gamma(1 - \pi_{t,i})}{\sqrt{\pi_{t,i}}}\right)^2 \frac{1}{\pi_{t,i}} + \frac{(\eta_t \xi_t)^2}{\pi_{t,i}^2} - 2\left(\ell_{t,i} - \frac{\gamma(1 - \pi_{t,i})}{\sqrt{\pi_{t,i}}}\right) \frac{\eta_t \xi_t}{\sqrt{\pi_{t,i}}}\right]$$

$$= \mathbb{E}_t\left[\left(\left(\ell_{t,i} - \frac{\eta_t(\xi_t + \gamma(1 - \pi_{t,i}))}{\sqrt{\pi_{t,i}}}\right) \frac{\mathbb{I}(A_t = i)}{\pi_{t,i}}\right)^2\right]$$

**Assumption 1.** *Assume that for all $t \in [T], i \in [K]$ it holds that $|\eta_t \sqrt{\pi_{t,i} \hat{L}_{t,i}}| \leq \frac{1}{6}$.*

*Lemma 13.* First for non-negative $\ell$ we show that $\epsilon \in [0, \ell]$. This follows by observing that $1/(1 + 3/2x)^2 \leq 1 - 2x$ for $x \in [0, 1/6]$ and $1/(1 + x)^2 \geq 1 - 2x$ for $x \geq 0$ so by the Intermediate Value theorem there exists an $\epsilon \in [0, \ell]$ such that equality is obtained. Next, for $\ell < 0$ for $\epsilon = 0$ we have $1/(1 - x)^2 \geq 1 + 2x$ for $x \geq 0$ and for $\epsilon = \ell/2$ we have $1/(1 - x/2)^2 \leq 1 - 2x$ for $x \in [0, 1/4]$ and so we have $\epsilon = [-\frac{|\ell|}{2}, \frac{|\ell|}{2}]$.

For the second part of the lemma using the Taylor expansion around 0 of $1/(1 + x)^2$ implies that

$$\frac{1}{(1 + \eta\sqrt{\pi}\tilde{\ell})^2} \leq 1 - 2\eta\sqrt{\pi}\tilde{\ell} + 3(\eta\sqrt{\pi}\tilde{\ell})^2,$$

and so

$$1 - 2\eta\sqrt{\pi}(\tilde{\ell} - \epsilon) \leq 1 - 2\eta\sqrt{\pi}\tilde{\ell} + 3(\eta\sqrt{\pi}\tilde{\ell})^2 \iff$$
$$2\eta\sqrt{\pi}\epsilon \leq 3(\eta\sqrt{\pi}\tilde{\ell})^2 \iff$$
$$2\eta\sqrt{\pi}\epsilon \leq 3(\eta\sqrt{\pi}(\ell + \epsilon))^2 \implies$$
$$\epsilon \leq 4\eta\sqrt{\pi}\ell^2.$$

$\square$

**Stochastic bound.** Let $D_t^{TS}(u, w) = \frac{1}{\eta_t} D_{TS}(u, v)$ and let

$$\tilde{\pi}_{t+1,i} = \frac{\pi_{t,i}}{(1 + \eta_{t+1}\sqrt{\pi_{t,i}}(\hat{\ell}_{t,i} + \epsilon_{t,i} - \lambda_t))^2},$$

where $\lambda_t = \ell_{t,A_t}$. We note that $\pi_{t+1}$ is now the projection of $\tilde{\pi}_{t+1}$ onto the simplex. Further by the 3-point rule for Bregman divergence we have that

$$\langle \hat{L}_t, \pi_t - u \rangle = D_t^{TS}(u, \pi_t) - D_t^{TS}(u, \tilde{\pi}_{t+1}) + D_t^{TS}(\pi_t, \tilde{\pi}_{t+1})$$
$$\leq D_t^{TS}(u, \pi_t) - D_t^{TS}(u, \pi_{t+1}) + D_t^{TS}(\pi_t, \tilde{\pi}_{t+1}).$$

**Penalty term.**

**Lemma 16** (Lemma 14). *For stochastic losses the penalty term is bounded as follows*

$$\mathbb{E}[D_{t+1}^{TS}(u, \pi_{t+1}) - D_t^{TS}(u, \pi_{t+1})] \leq O\left(\frac{\mathbb{E}\left[\left(\sum_{i \neq i^*} \pi_{t+1,i}\right)^2\right] \sqrt{K} \log(t)}{\sqrt{t}}\right),$$

*where $D_t^{TS}(u, v) = \frac{1}{\eta_t} D_{TS}(u, v)$ and $\eta_t = \frac{1}{\sqrt{t}}$.*

*Proof.* for $u = e_{i^*}$:

$$D_{t+1}^{TS}(u, \pi_{t+1}) - D_t^{TS}(u, \pi_{t+1}) = -\left(\frac{1}{\eta_{t+1}} - \frac{1}{\eta_t}\right)$$
$$+ \left(\frac{1}{\eta_{t+1}} - \frac{1}{\eta_t}\right)\left(\frac{1}{\sqrt{\pi_{t+1,i^*}}} - 1\right)$$
$$+ \left(\frac{1}{\eta_{t+1}} - \frac{1}{\eta_t}\right)\sum_{i=1}^{K}\sqrt{\pi_{t+1,i}}$$
$$= \left(\frac{1}{\eta_{t+1}} - \frac{1}{\eta_t}\right)\frac{1 - \sqrt{\pi_{t+1,i^*}} + \pi_{t+1,i^*}}{\sqrt{\pi_{t+1,i^*}}}$$
$$+ \left(\frac{1}{\eta_{t+1}} - \frac{1}{\eta_t}\right)\sum_{i \neq i^*}\sqrt{\pi_{t+1,i}}$$
$$= \left(\frac{1}{\eta_{t+1}} - \frac{1}{\eta_t}\right)\frac{(1 - \sqrt{\pi_{t+1,i^*}})^2}{\sqrt{\pi_{t+1,i^*}}}$$
$$+ \left(\frac{1}{\eta_{t+1}} - \frac{1}{\eta_t}\right)\sum_{i \neq i^*}\sqrt{\pi_{t+1,i}}.$$

From the update in Equation 4 we have

$$\frac{1}{\sqrt{\pi_{t+1,i^*}}} = \sqrt{K} + \sum_{s=1}^{t}\eta_s(\hat{L}_{s,i^*} + \epsilon_{s,i^*}),$$

which implies

$$\frac{(1 - \sqrt{\pi_{t+1,i^*}})^2}{\sqrt{\pi_{t+1,i^*}}} \leq \sqrt{K}(1 - \sqrt{\pi_{t+1,i^*}})^2 + (1 - \sqrt{\pi_{t+1,i^*}})^2\sum_{s=1}^{t}\eta_s(\hat{L}_{s,i^*} + \epsilon_{s,i^*})$$

First we bound $(1 - \sqrt{\pi_{t+1,i^*}})^2$:

$$(1 - \sqrt{\pi_{t+1,i^*}})^2 = \left(1 - \sqrt{1 - \sum_{i \neq i^*}\pi_{t+1,i}}\right)^2 = \left(\frac{\sum_{i \neq i^*}\pi_{t+1,i}}{1 + \sqrt{1 - \sum_{i \neq i^*}\pi_{t+1,i}}}\right)^2$$
$$\leq \left(\sum_{i \neq i^*}\pi_{t+1,i}\right)^2.$$

In the stochastic case WLOG we can take $\mathbb{E}[\ell_{t,i^*}] = 0, \forall t \in [T]$. We first control $\hat{L}_{t,i^*}$. We have

$$\hat{L}_{t,i^*} = \left(\ell_{t,i^*} - \frac{\eta_t\gamma(1 - \pi_{t,i^*})}{\sqrt{\pi_{t,i^*}}}\right)\frac{\mathbb{I}(A_t = i^*)}{\pi_{t,i^*}} - \frac{\eta_t\xi_t}{\sqrt{\pi_{t,i^*}}} - \sum_{i=1}^{K}\frac{\pi_{t,i}^{\frac{3}{2}}\hat{\ell}_{t,i}}{\sum_{j=1}^{K}\pi_{t,j}^{\frac{3}{2}}}$$

The first term, $\ell_{t,i^*}\frac{\mathbb{I}(A_t=i^*)}{\pi_{t,i^*}}$ is 0 in expectation. The second term, $-\frac{\eta_t\gamma(1-\pi_{t,i^*})}{\sqrt{\pi_{t,i^*}}}\frac{\mathbb{I}(A_t=i^*)}{\pi_{t,i^*}}$, will be used to cancel out the contribution from the perturbation $\epsilon_{t,i^*}$. The third term is there to help with the adversarial setting analysis. Next, we decompose the fourth term as

$$\sum_{i=1}^{K}\frac{\pi_{t,i}^{\frac{3}{2}}\hat{\ell}_{t,i}}{\sum_{j=1}^{K}\pi_{t,j}^{\frac{3}{2}}} = \frac{1}{\sum_{j=1}^{K}\pi_{t,j}^{3/2}}\sum_{i=1}^{K}\sqrt{\pi_{t,i}}\ell_{t,i}\mathbb{I}(A_t = i) - \eta_t\gamma(1 - \pi_{t,i})\mathbb{I}(A_t = i) - \pi_{t,i}\eta_t\xi_t.$$

The first part of the above has a non-positive contribution to $\hat{L}_{t,i^*}$ in expectation. The only non-negative contribution now comes from

$$\frac{1}{\sum_{j=1}^{K}\pi_{t,j}^{3/2}}\sum_{i=1}^{K}\eta_t(\xi_t + \gamma(1 - \pi_{t,i}))\mathbb{I}(A_t = i) \leq \sqrt{K}(\eta_t\xi_t + \eta_t\gamma) \leq 2\sqrt{K}\eta_t\gamma$$

and so we bound

$$\sum_{s=1}^{t} -\eta_s \sum_{i=1}^{K} \frac{\pi_{s,i}^{\frac{3}{2}} \hat{\ell}_{s,i}}{\sum_{j=1}^{K} \pi_{s,j}^{\frac{3}{2}}} \leq \sum_{s=1}^{t} 2\eta_s \sqrt{K} \eta_s \gamma \leq 64\sqrt{K} \log(t).$$

Next we are going to bound $\mathbb{E}[\epsilon_{t,i^*}]$ using Lemma 10 together with Lemma 13:

$$\mathbb{E}[\epsilon_{t,i^*}] \leq \mathbb{E}[\eta_t \sqrt{\pi_{t,i^*}} \hat{L}_{t,i^*}^2] \leq \frac{13}{2} \mathbb{E}[\eta_t (1 - \pi_{t,i^*})/\sqrt{\pi_{t,i^*}}].$$

This term is exactly canceled out by $\frac{\eta_t \gamma (1 - \pi_{t,i^*})}{\sqrt{\pi_{t,i^*}}}$ as $\gamma = \frac{13}{2}$. For the final bound we have

$$\mathbb{E}\left[\left(\frac{1}{\eta_{t+1}} - \frac{1}{\eta_t}\right) \frac{(1 - \sqrt{\pi_{t+1,i^*}})^2}{\sqrt{\pi_{t+1,i^*}}}\right]$$

$$\leq \left(\frac{1}{\eta_{t+1}} - \frac{1}{\eta_t}\right) \mathbb{E}\left[\left(\sum_{i \neq i^*} \pi_{t+1,i}\right)^2\right] + 4\left(\frac{1}{\eta_{t+1}} - \frac{1}{\eta_t}\right) \mathbb{E}\left[\left(\sum_{i \neq i^*} \pi_{t+1,i}\right)^2 \sum_{s=1}^{t} \eta_s(\hat{L}_{s,i^*} + \epsilon_{s,i^*})\right]$$

$$\leq \left(\frac{1}{\eta_{t+1}} - \frac{1}{\eta_t}\right) \mathbb{E}\left[\left(\sum_{i \neq i^*} \pi_{t+1,i}\right)^2\right] + \mathbb{E}\left[\left(\frac{1}{\eta_{t+1}} - \frac{1}{\eta_t}\right)(1 - \sqrt{\pi_{t+1,i^*}})^2 \sum_{s=1}^{t} -\eta_s \sum_{i=1}^{K} \frac{\pi_{s,i}^{\frac{3}{2}} \hat{\ell}_{s,i}}{\sum_{j=1}^{K} \pi_{s,j}^{\frac{3}{2}}}\right]$$

$$+ \mathbb{E}\left[\sum_{s=1}^{t} \eta_s \left(\epsilon_{s,i^*} - \frac{\gamma \eta_s (1 - \pi_{s,i^*})}{\sqrt{\pi_{s,i^*}}}\right)\right]$$

$$\leq 32 \frac{\mathbb{E}\left[\left(\sum_{i \neq i^*} \pi_{t+1,i}\right)^2\right] \sqrt{K} \log(t)}{\sqrt{t}}.$$

$\square$

**Stability term.** Recall that the stability term is $D_t^{TS}(\pi_t, \tilde{\pi}_{t+1})$. This term is bounded in a standard way. We proceed to do so as follows for any $t \geq 4\sqrt{K}$:

**Lemma 17** (Lemma 15). *For stochastic losses the stability term is bounded as follows*

$$\mathbb{E}[D_t^{TS}(\pi_t, \tilde{\pi}_{t+1})] \leq O\left(\frac{1}{\sqrt{t}} \sum_{i=1}^{K} \sqrt{\pi_{t,i}}(1 - \pi_{t,i}) + \frac{1}{t}\right),$$

*where* $D_t^{TS}(u, v) = \frac{1}{\eta_t} D_{TS}(u, v).$

*Proof.* We have the following

$$D_t^{TS}(\pi_t, \tilde{\pi}_{t+1}) = \frac{1}{\eta_t} \sum_{i=1}^{K} 2\sqrt{\pi_{t+1,i}} - 2\sqrt{\pi_{t,i}} - \frac{1}{\sqrt{\pi_{t+1,i}}}(\pi_{t+1,i} - \pi_{t,i})$$

$$= \frac{1}{\eta_t} \sum_{i=1}^{K} \sqrt{\pi_{t+1,i}} - 2\sqrt{\pi_{t,i}} + \sqrt{\pi_{t,i}}\left(1 + \eta_{t+1}\sqrt{\pi_{t,i}}(\hat{\ell}_{t,i} + \epsilon_{t,i} - \lambda_t)\right)$$

$$= \frac{1}{\eta_t} \sum_{i=1}^{K} \sqrt{\pi_{t+1,i}} - \sqrt{\pi_{t,i}} + \eta_{t+1}\pi_{t,i}(\hat{\ell}_{t,i} + \epsilon_{t,i} - \lambda_t)$$

$$= \frac{1}{\eta_t} \sum_{i=1}^{K} \sqrt{\pi_{t,i}}\left(\frac{1}{1 + \eta_{t+1}\sqrt{\pi_{t,i}}(\hat{\ell}_{t,i} + \epsilon_{t,i} - \lambda_t)} - 1\right) + \eta_{t+1}\pi_{t,i}(\hat{\ell}_{t,i} + \epsilon_{t,i} - \lambda_t)$$

$$\leq \frac{1}{\eta_t} \sum_{i=1}^{K} \sqrt{\pi_{t,i}}\left(-\eta_{t+1}\sqrt{\pi_{t,i}}(\hat{\ell}_{t,i} + \epsilon_{t,i} - \lambda_t) + 2\eta_{t+1}^2\pi_{t,i}(\hat{\ell}_{t,i} + \epsilon_{t,i} - \lambda_t)^2\right)$$

$$+ \eta_{t+1}\pi_{t,i}(\hat{\ell}_{t,i} + \epsilon_{t,i} - \lambda_t) \qquad\qquad (\tfrac{1}{1+x} \leq 1 - x + 2x^2 \text{ for } x \geq -\tfrac{1}{2})$$

$$\leq 2\eta_t \sum_{i=1}^{K} \pi_{t,i}^{3/2}(\hat{\ell}_{t,i} + \epsilon_{t,i} - \lambda_t)^2,$$

where for the second to last inequality we only need to check $\eta_{t+1}\sqrt{\pi_{t,i}}(\epsilon_{t,i} - \lambda_t) \geq -\frac{1}{2}$. We have $\eta_{t+1}\sqrt{\pi_{t,i}}\lambda_t \geq -\frac{1}{\sqrt{t}}$ and Lemma 13 implies

$$\mathbb{E}[\eta_{t+1}\sqrt{\pi_{t,i}}\epsilon_{t,i}] \geq -\eta_t^2 \,\mathbb{E}[\pi_{t,i}|\hat{L}_{t,i}|^2] \geq -\Omega(\tfrac{1}{t}).$$

We bound $\mathbb{E}[\pi_{t,i}^{3/2}(\hat{\ell}_{t,i} + \epsilon_{t,i} - \lambda_t)^2] \leq 2\,\mathbb{E}[\pi_{t,i}^{3/2}\epsilon_{t,i}^2] + 2\,\mathbb{E}[\eta_{t+1}^2\pi_{t,i}^{3/2}(\hat{\ell}_{t,i} - \lambda_t)^2]$. For the first term we have

$$2\,\mathbb{E}[\eta_t^2\pi_{t,i}^{5/2}|\hat{L}_{t,i}|^4] \leq O\left(\frac{1}{\sqrt{t}}\right).$$

For the second term we use Lemma 10 to get $\mathbb{E}[\eta_{t+1}^2\pi_{t,i}^{3/2}(\hat{\ell}_{t,i} - \lambda_t)^2] \leq \frac{13}{2}\,\mathbb{E}[\eta_{t+1}^2\sqrt{\pi_{t,i}}(1 - \pi_{t,i})]$. $\qquad\square$

### Self-bounding the regret for stochastic losses.

*Theorem 4, stochastic losses.* Combining the bound in Lemma 14 and Lemma 15 together with the adversarial bound we have that the total regret is bounded as follows

$$+ O\left(\sum_{t=T_0}^{T} \sum_{i\neq i^*} \sqrt{\pi_{t,i}}\left(\frac{1}{\sqrt{t}} - \sqrt{\pi_{t,i}}\left(\Delta_i - \frac{\sqrt{K}\log(t)}{\sqrt{t}}\right)\right) + \sum_{t=1}^{T_0-1} \frac{\pi_{t,i}\sqrt{K}\log(t)}{\sqrt{t}}\right)$$

$$+ O\left(K^{3/2}\right)$$

In the above we bound the lower order term from the stability as $\sum_{t=1}^{T}\sum_{i=1}^{K} \gamma_t^2\sqrt{\pi_{t,i}} = O(K^{3/2})$ and decompose the regret into four parts. The first and second line correspond to the two terms from the penalty bound. Each of the two lines are decomposed into two terms. The first term is the result of the self-bounding trick and the second term is the additional regret for the initial number of rounds before the self-bounding trick can be applied.

We repeatedly use the following inequality $2a\sqrt{x} - bx \leq \frac{a^2}{b}$, which holds for $a, b \geq 0$. For the first line of the decomposition we take $T_0 = 8\frac{K\log^2(K/\Delta_{min})}{\Delta_{min}}$, where $\Delta_{min}$ is the smallest non-zero expected loss. We note that

$$\frac{\sqrt{K}\log(T_0)}{\sqrt{T_0}} = \Delta_{min}\frac{\log(8K\log^2(1/\Delta_{min}))}{8\log(K/\Delta_{min})} = \Delta_{min}\left(\frac{\log(K)}{8\log(K/\Delta)} + \frac{\log(16\log(1/\Delta_{min}))}{8\log(K/\Delta_{min})}\right) \leq \frac{\Delta_{min}}{2},$$

for any $\Delta_{min} \leq \frac{1}{2024}$. If $\Delta_{min} > \frac{1}{2024}$, we take $T_0 = 8\frac{K\log^2(2024)}{\Delta_{min}}$. The above implies that $\sqrt{\pi_{t,i}}\left(\frac{1}{\sqrt{t}} - \sqrt{\pi_{t,i}}\left(\Delta_i - \frac{\sqrt{K}\log(t)}{\sqrt{t}}\right)\right) \leq \frac{2}{t\Delta_i}$ and further $\sum_{i\neq i^*}\sum_{t=1}^{T_0-1}\frac{\pi_{t,i}\sqrt{K}\log(t)}{\sqrt{t}} = O(\frac{K\log^2(1/\Delta_{\min})}{\Delta_{\min}})$. The final regret bound is

$$O\left(\sum_{i\neq i^*}\frac{\log(T)}{\Delta_i} + \frac{K\log^2(1/\Delta_{\min})}{\Delta_{\min}} + K^{3/2}\right).$$

$\square$

**Adversarial losses.** We now present the argument for the regret bound in the adversarial setting.

*Theorem 4, adversarial losses.* Let $\tilde{\ell}_{t,i} = \left(\ell_{t,i} - \frac{\eta_t\gamma(1-\pi_{t,i})}{\sqrt{\pi_{t,i}}}\right)\frac{\mathbb{I}(A_t=i)}{\pi_{t,i}}$ and $\chi_t = \frac{\eta_t}{\eta_{t-1}}$ We recall the update for Algorithm 2 in Fang et al. [2022]

$$\hat{w}_{t+1} = \nabla\Phi(\pi_t) - \eta_{t-1}(\tilde{\ell}_t + \epsilon_t),$$
$$\hat{y}_{t+1} = \chi_t\hat{w}_{t+1} + (1-\chi_t)\nabla\Phi(\pi_1),$$
$$\pi_{t+1} = \nabla\Phi^*(\hat{y}_{t+1}),$$

where $\Phi$ is the $1/2$-Tsallis potential plus the indicator function for the probability simplex $\Delta^{K-1}$. Since $\pi_1$ is uniform the second step of the update is equivalent to $\hat{y}_{t+1} = \chi_t\hat{w}_{t+1}$. Re-writing the first step of the update we have

$$-\hat{y}_{t+1,i} = \frac{\chi_t}{\sqrt{\pi_{t,i}}} + \eta_t(\tilde{\ell}_{t,i} + \epsilon_{t,i}) = \frac{1}{\sqrt{\pi_{t,i}}} + \eta_{t+1}(\tilde{\ell}_{t,i} + \epsilon_{t,i}) - \eta_t\left(\frac{1}{\eta_t} - \frac{1}{\eta_{t-1}}\right)\frac{1}{\sqrt{\pi_{t,i}}}$$

$$= \frac{1}{\sqrt{\pi_{t,i}}} + \eta_t\left(\left(\ell_{t,i} - \frac{\eta_t\gamma(1-\pi_{t,i})}{\sqrt{\pi_{t,i}}}\right)\frac{\mathbb{I}(A_t=i)}{\pi_{t,i}} + \epsilon_{t,i} - \left(\frac{1}{\eta_t} - \frac{1}{\eta_{t-1}}\right)\frac{1}{\sqrt{\pi_{t,i}}}\right)$$

$$= \frac{1}{\sqrt{\pi_{t,i}}} + \eta_t(\hat{\ell}_{t,i} + \lambda_t).$$

Since $\nabla\Phi^*$ is invariant under constant vector perturbations we finally have

$$\pi_{t+1,i} = \nabla\Phi^*(\hat{y}_{t+1})_i = \nabla\Phi^*\left(-\frac{1}{\sqrt{\pi_t}} - \eta_t(\hat{\ell}_{t+1} + \epsilon_t)\right) = \nabla\Phi^*\left(-\frac{1}{\sqrt{\pi_t}} - \eta_t(\hat{L}_{t+1} + \epsilon_t)\right)$$

$$= \frac{1}{(1/\sqrt{\pi_{t,i}} + \eta_t(\hat{L}_{t,i} + \epsilon_{t,i}))^2}$$

$$= \frac{\pi_{t,i}}{(1 + \eta_t\sqrt{\pi_{t,i}}(\hat{L}_{t,i} + \epsilon_{t,i}))^2}.$$

And so the update in Algorithm 2 of Fang et al. [2022] is equivalent to the perturbed OMD update which we have shown enjoys an optimistic regret guarantee. The regret guarantee in the adversarial setting is now recovered from Theorem 3 in Fang et al. [2022]. In particular the theorem guarantees that the regret is bounded as

$$\sum_{t=1}^T D_t^{TS}(\pi_t, \nabla F^*(\nabla F(\pi_t) - \eta_t(\hat{\ell}_t + \epsilon_t))) + \sqrt{KT}.$$

Every term in the sum is bounded in the same way as the stability terms, that is $D_t^{TS}(\pi_t, \nabla F^*(\nabla F(\pi_t) - \eta_t(\hat{\ell}_t + \epsilon_t))) \leq \sum_{i=1}^K \pi_{t,i}^{3/2}(\hat{\ell}_{t,i}^2 + \epsilon_{t,i})$. We can now use the bound in the proof of Lemma 15 to complete proof of the adversarial bound and the proof of Theorem 4. $\square$

