# OpenReview forum: "PRODuctive bandits: Importance Weighting No More"
_NeurIPS.cc/2024/Conference — NeurIPS 2024 poster_

### Official Review · Reviewer_1VVH · 2024-07-09

**Soundness:** 4
**Presentation:** 3
**Contribution:** 4
**Rating:** 7
**Confidence:** 4

**Summary:**

This paper first considers incentive-compatible online learning problem, which is first considered in [Freeman et al., 2020]. Specifically, the original WSU-UX algorithm only achieves $O(T^{2/3})$ regret shown in [Freeman et al., 2020]. By injecting a bias in the loss estimator constuction to the original WSU-UX algorithm, the authors achieve $O(\sqrt{KT\log K})$ regret for this problem. The added biases can be viewed as certain second-order approximation of the orginal exponential weight algorithm. Besides this algorithm, the authors also propose another algorithm which is still incentive-compatible and interestingly does not use the inverse propensity weighting. The authors show that the LB-Prod algorithm is similar to FTRL with log-barrier regularizer and also achieves $O(\sqrt{KT\log T})$ optimal regret. The authors also consider the Tsallis entropy variant of this Prod-type algorithm and achieve best-of-both-world guarantee for multi-armed bandits in adversarial and stochastic environments.

**Strengths:**

- This problem resolves an open problem proposed in [Freeman et al., 2020], achieving $O(\sqrt{T})$-type regret guarantee for incentive-compatible online learning with bandit feedback. The proposed two Prod-type algorithms in the bandit feedback setting are very interesting to me. The relationship between LB-Prod and FTRL with log-barrier regularizer shown in the analysis is also new to me.

- The proposed Tsallis variant of Prod algorithm is also interesting since most previous BOBW algorithm relies on the FTRL framework.

**Weaknesses:**

I do not find main weakness of this paper but I have some concerns on the results and some configurations of the algorithms.

- While the authors argue that the added bias in the first algorithm is related to the second-order approximation of the exponential weight algorithm, I still have difficulty understanding why the bias is of this form.

- In Section "perturbation analysis", the authors argue that there is an equivalence between LB-Prod and LB-FTRL with a shifted loss. I wonder whether the standard OMD analysis also works? This can make the relationship between LB-Prod and LB-FTRL clearer.

- The BOBW regret for the stochastic environment is a bit away from the optimal due to the additional $K/\Delta_{\min}$ lower order term.

**Questions:**

- Can the authors explain more on what difficulties are if one uses OMD with Logbarrier and perturbation analysis to analyze LB-Prod?

- Whether there is also an inverse-propensity-weighting-free version of Tsallis-Prod that achieves BOBW for adversarial MAB?

- Can the authors explain more about how the added bias in the modification of WSU-UX algorithm is related to the second-order approximation of MWU? Does this similarity come from the algorithm dynamic or the analysis perspective?

- Can the stochastic regret bound for Tsallis-Prod be improved to optimal?

**Limitations:**

See "weakness" and "questions".

---

> ### Author Rebuttal · Authors · 2024-08-06
>
> We address weaknesses and questions below:
>
> Weaknesses:
>
> --Difficulty understanding the form of the bias: The exponentiated weights/hedge algorithm update can be written as the first equality after line 149, where $\lambda_t$ is the normalization factor which makes $\pi_{t+1}$ a probability distribution. The second order approximation intuitively replaces the exponential by its second order Taylor expansion. That is we exactly write the exponential as a Taylor series and then just truncate to the quadratic term.
>
> --Section "perturbation analysis": We believe that the reviewer means there is an equivalence between LB-Prod and LB-OMD. There indeed is such an equivalence and one can use the standard OMD analysis, however, it would be on a perturbed version of the losses. It turns out we can control these perturbations so the analysis will go through. We analyze a version of TS-Prod precisely in this way in Appendix D.
>
> --BoBW regret: The lower order term $\frac{K}{\Delta_{\min}}$ only shows up in the analysis for the algorithm presented in Appendix D and we believe this to be a shortcoming of our analysis. The term does not show up for the algorithm presented in Section 5 with regret bound in Theorem 3.
>
> Questions:
> >Can the authors explain more on what difficulties are if one uses OMD with Logbarrier and perturbation analysis to analyze LB-Prod?
>
> The analysis is not significantly more difficult. We refer the reviewer to Appendix D, Lemma 13 for an idea of how one can bound the perturbations. This lemma is stated for a variant of TS-Prod, however, a similar result can be derived for LB-Prod.
>
> >Whether there is also an inverse-propensity-weighting-free version of Tsallis-Prod that achieves BOBW for adversarial MAB?
>
> We are not sure that this is possible. We have heavily used the form of the log-barrier potential update when deriving the result for LB-Prod.
>
> >Can the authors explain more about how the added bias in the modification of WSU-UX algorithm is related to the second-order approximation of MWU? Does this similarity come from the algorithm dynamic or the analysis perspective?
>
> See our answer in the weaknesses part of the rebuttal.
>
> >Can the stochastic regret bound for Tsallis-Prod be improved to optimal?
>
> The bound is already asymptotically optimal. We note that our analysis does not tackle the multiple best arms setting, however, we believe that the work of Ito 2021 can be used to derive regret bounds in this setting as well, see our response to reviewer *rzG8* as well. The only other sub-optimality of the presented bound in Theorem 3 is in terms of some constant scalar factors which could potentially be tightened.

---

> > ### Comment · Reviewer_1VVH · 2024-08-10
> >
> > I thank the authors for their response and I keep my positive score.

---

### Official Review · Reviewer_r5go · 2024-07-09

**Soundness:** 3
**Presentation:** 2
**Contribution:** 4
**Rating:** 7
**Confidence:** 3

**Summary:**

This paper revisits the simple PROD-type algorithms, originally introduced by Cesa-Bianchi et al. in 2007 for online learning under full feedback, and does an excellent job in doing so. Specifically, the authors demonstrate that some version of these algorithms can achieve optimality even for the $K$-armed bandit problem. They show that three different types of prod algorithms can achieve optimality. Interestingly, one of these algorithms can be seen as the first multi-armed bandit algorithm that solves the problem and does not rely on importance weighting. Notably, one of these algorithms also enjoys best-of-both-worlds guarantees. As another significant corollary of their work, the paper refutes a conjecture by Freeman et al. (2020) about incentive-compatible multi-armed bandits being intrinsically more difficult than regular multi-armed bandits.

**Strengths:**

1) The paper provides valuable insights about how to generate PROD-type algorithms as first-order OMD approximation with biased losses of classic multi-armed bandits algorithms.
2) The paper provides the first importance-weighting free algorithm to solve the multi-armed bandits problem.
3) The paper solves (negatively) a conjecture about incentive-compatible multi-armed bandits being more difficult than ordinary multi-armed bandits.
4) The paper provides a PROD-type algorithm that enjoys BOBW guarantees.

**Weaknesses:**

The presentation targets a specialized audience, assuming the reader is well-versed in template proofs of OMD (Online Mirror Descent) and FTRL (Follow-The-Regularized-Leader) algorithms, as well as techniques such as "change of measure".
To enhance accessibility for less experienced readers, consider adding dedicated appendices or providing precise references to relevant literature where these templates/definitions/techniques can be found. These additions can help guide readers through the nuances of the paper, broadening the potential audience of the paper, which I believe deserves to be broad.

**Questions:**

Do you believe that your techniques can be applied to other problems, such as contextual/linear/combinatorial bandits, or to feedback graph and partial monitoring? Do you foresee any challenges in adapting these techniques beyond the domain of classic multi-armed bandits?

**Limitations:**

I believe that this category does not apply to a paper which is entirely theoretical.

---

> ### Author Rebuttal · Authors · 2024-08-06
>
> We address weaknesses and questions below:
>
> Weaknesses:
>
> --The presentation targets a specialized audience: Thank you for the suggestion. We agree that the presentation in the main paper is quite technical. We are happy to add an appendix which describes the standard analysis of OMD and gives a quick overview of “change of measure” trick. Further, for a more thorough example of the change of measure trick we will point the reader to the works of *Dylan J. Foster, Claudio Gentile, Mehryar Mohri, Julian Zimmert. Adapting to Misspecification in Contextual Bandits* and *Haipeng Luo, Chen-Yu Wei, Chung-Wei Lee. Policy Optimization in Adversarial MDPs: Improved Exploration via Dilated Bonuses*.
>
> Questions:
> > Do you believe that your techniques can be applied to other problems, such as contextual/linear/combinatorial bandits, or to feedback graph and partial monitoring? Do you foresee any challenges in adapting these techniques beyond the domain of classic multi-armed bandits?
>
> With regards to the online learning with graph feedback setting we expect that the results will transfer over in a somewhat straightforward way, at least for the adversarial setting. With regards to the contextual/linear bandit setting it is perhaps possible to come up with a similar update for log-barrier type algorithms. One challenging aspect might be extending our results to small loss bounds due to the way we bias the rewards in WSU-UX, see also our response to reviewer **rzG8**.

---

### Official Review · Reviewer_zjT8 · 2024-07-13

**Soundness:** 2
**Presentation:** 2
**Contribution:** 2
**Rating:** 5
**Confidence:** 3

**Summary:**

Adversarial bandits have typically been solved through techniques such as online mirror descent or variants of multiplicative weight updates, but these might rely on some notion of importance-based weighting. To develop solutions free of such importance-weighting, the authors propose using the Prod algorithm, which serves as a simpler approximation of online mirror descent. Algorithms based on Prod can then be combined with Tsalis regularization to get a best-of-both-worlds guarantee, in both the adversarial and stochastic regimes.

**Strengths:**

1. **Paper resolves open question** - As noted in the discussion in Section 6, the authors resolve an open question related to incentive compatibility and bandits posed by prior work. By resolving such a conjecture, they're able to provide a bridge between full-information and bandit settings.
2. **Algorithm provides nice best-of-both-world guarantee** - Section 5 of the paper introduces an application of the Prod-based methodology using the Tsallis regularizer, and shows how this can achieve a best-of-both-worlds guarantee. While the connection with the previous sections, and the overall story of the paper, isn't super clear, the bounds proven in Section 5 provide some nice properties when using Prod.

**Weaknesses:**

1. **Unclear rationale behind using Prod** - Throughout the paper, the Prod algorithm is introduced and detailed, yet it is unclear why such an algorithm is necessary and the types of benefits brought upon by using such an algorithm. Early in the paper, the authors suggest that the simplicity of their update rules provides motivations for using Prod, but this motivation is not expanded upon, making it difficult to understand the motivation of the paper.
2. **Unclear why importance weighting is undesirable** - Another rationale pointed out throughout the paper is that Prod can eliminate the need for performing importance weighting. However, it's unclear why such a property is desirable. The paper never details why we should eliminate importance weighting, and what the benefits are for algorithms that lack this.
3. **Connection between Section 4 and 5 is unclear** - Section 4 of the paper details how Prod leads to a lack of importance weighting, while Section 5 details the best-of-both-worlds property with Prod. While both sections are interesting on their own, it is unclear how they each contribute to the main idea of the story. From my understanding, it seems to be the lack of a need to use importance weighting. In this case, it is not clear how Section 5 contributes to that story, and how Section 5 builds on the ideas from Section 4.

**Questions:**

1. What is the benefit of using Prod/what is the rationale for studying this problem?

**Limitations:**

The paper lacks any limitations section. The authors write that the paper lacks a limitations section because it is "purely theoretical work", though I believe that limitations in the analysis or assumptions made should be pointed out even for theoretical work.

---

> ### Author Rebuttal · Authors · 2024-08-06
>
> We address weaknesses and questions below:
>
> Weaknesses:
> --Unclear rationale behind using Prod: As we have already stated the Prod updates are simple and closed form, that is they do not require solving an optimization problem at every step. Prod type algorithms have been previously used for Online Learning problems but they were not known to be optimal in the partial information setting. As we have pointed out in the paper, there was even a conjecture that Prod style updates can not achieve the optimal min-max regret in the adversarial setting. We think that this alone makes Prod style algorithms interesting enough to study. Further, we are able to show the **first** importance weighting free algorithm for bandits which is also a very interesting contribution by itself. Finally, this specific type of Prod updates allows for solving the incentive compatible bandit problem studied by Freeman et al. 2020
>
> --Unclear why importance weighting is undesirable: The benefit is simple and we have already highlighted that in lines 161-162 in Section 4. To reiterate, importance weight free algorithms do not need to control the players probabilities in any way. Further, such algorithms will only have to deal with bounded losses.
>
> --Connection between Section 4 and 5 is unclear: The best-of-both worlds problem has been well motivated and studied, see *Sebastien Bubeck and Aleksandrs Slivkins. The best of both worlds: Stochastic and adversarial bandits* and *Zimmert and Seldin. Tsallis-INF: An Optimal Algorithm for Stochastic and Adversarial Bandits*. Whenever a new bandit algorithm is introduced it is natural to ask if that algorithm will perform well in both stochastic and adversarial settings.
>
> Questions:
> >What is the benefit of using Prod/what is the rationale for studying this problem?
>
> See our answers above.
>
> Limitations:
> >The paper lacks any limitations section...
>
> First, we want to point out that we have stated all assumptions clearly. Second, we believe our theoretical results do not hide any limitations of the algorithms and are stated in a formal and sound way. We welcome any further feedback about what limitations we have failed to address in terms of our theory. The reviewer has given a score of 2 for the soundness of our results. We would like to address any soundness concerns and welcome any feedback on which part of our paper is not sound.

---

> > ### Comment · Reviewer_zjT8 · 2024-08-07
> > **Thank you for the comments**
> >
> > Thank you for your comments and information. A discussion of limitations would be helpful for practitioners aiming to leverage the algorithms from this paper. While I am still not convinced about the motivation for Prod updates, the rebuttal clarifies the connection between this and prior works, so I raise my score.

---

### Official Review · Reviewer_rzG8 · 2024-07-13

**Soundness:** 3
**Presentation:** 3
**Contribution:** 4
**Rating:** 7
**Confidence:** 3

**Summary:**

This paper presents novel algorithms for the multi-armed bandit (MAB) problem that are shown to achieve optimal regret bounds even without relying on importance weighting. The authors introduce variants of the well-known Prod algorithm that are effective for both adversarial and stochastic settings, presenting significant improvements over existing state-of-the-art methods. Main contributions include disproving the conjecture that Prod-like algorithms are sub-optimal for bandits (that is, bandit feedback is shown not to be much harder than full information for algorithms in the Prod family), introducing a variant with nearly optimal regret, and achieving best-of-both-worlds guarantees with logarithmic regret in stochastic regimes.

**Strengths:**

The main results of achieving optimal regret bounds with variants of Prod and even optimal best-of-both-worlds guarantees are quite surprising and interesting. Even if the update rule of the proposed algorithm is extremely simple, this is shown to have a controlled effect on the regret via careful corrections.
Additionally, the paper is nicely structured and smoothly introduces the variants of Prod by providing intuitive reasoning behind their design.

The techniques adopted, together with the intuitions and the analogies provided, are also nontrivial and require a novel analysis template that has the potential to lead to simple algorithms with near-optimal guarantees for other online learning problems.

**Weaknesses:**

I have no major weaknesses to report. On a side note, having more details on the setting of incentive-compatible online learning would have been better, as it is a central motivation for this work. For instance, even just explicitly defining what a “proper scoring rule” is and more technical motivations as to why Prod-like algorithms are important in this setting.

**Questions:**

- For the stochastic regret guarantee of TS-Prod, do you require that the optimal arm is unique? If so, please be explicit about this assumption.
- At lines 155-159, you state that an overcorrection is necessary. However, a correction term of order $\\eta/\\tilde{\\pi}\_{t,i}$ instead of $\\eta\\ell\_{t,i} / \\tilde{\\pi}\_{t,i}$ could be one of the main factors that are preventing the achievement of first-order and second-order bounds. How much do you believe your current techniques for designing Prod-like algorithms prohibit achieving these guarantees?

Minor comments/typos:
- Line 7: “best-of-both-worlds” instead of “best-both-worlds”
- Table 1: the first column is simply the negative entropy, and calling it “KL divergence” too might be confusing
- Math display below line 130: the second equality should be an inequality if “Reg” is just an upper bound on the regret with the biased losses
- Line 172: for example, $T > K \\log K$ should suffice in order to satisfy the condition $T > (K/2) \\log T$ (the latter is not an explicit condition on $T$).
- Line 180: “be” instead of “be be”
- Line 211: “poses” instead of “posses”
- Line 214: “a fixed” instead of “a a fixed”
- Line 227: in “intermediate potential between KL and Logbarrier”, negative entropy is the potential function (whose Bregman divergence is the KL)
- Math display below line 360: $\\pi\_{t,i}^2$ between parentheses after the first equality should be $\\pi\_{t,i}$

**Limitations:**

The authors addressed potential limitations.

---

> ### Author Rebuttal · Authors · 2024-08-06
>
> We address weaknesses and questions below:
>
> Weaknesses:
>
> --More details on the setting of incentive-compatible online learning: We are happy to include more details for the incentive-compatible setting. We propose adding a quick overview which includes what constitutes a proper scoring rule together with extra motivation for why Prod algorithms are important for this setting in the main text and an extended discussion in the appendix. In fact a previous version of this work had a more careful discussion about the incentive-compatible framework and its motivation.
>
> Questions:
> >For the stochastic regret guarantee of TS-Prod, do you require that the optimal arm is unique? If so, please be explicit about this assumption.
>
> You are correct that our current analysis requires uniqueness of the optimal arm. We will state this explicitly. We expect that the techniques of Ito 2021 can be extended to our analysis which would allow for multiple optimal arms.
>
> Ito, Shinji. "Parameter-free multi-armed bandit algorithms with hybrid data-dependent regret bounds." Conference on Learning Theory. PMLR, 2021.
>
> >At lines 155-159, you state that an overcorrection is necessary...
>
> This is a great question. The analysis for WSU-UX will also go through with the smaller bias of $\eta \ell_{t,i}/\pi_{t,i}$. The reason we went with the larger bias is to keep the update linear in the losses which is important for the incentive compatible setting. However, we are not entirely sure if this is sufficient for small-loss bounds such as from Allenberg et al. 2006 (GREEN algorithm) because of the following. A quick calculation shows that the perturbation will contribute an additional term of $\eta\sum_{t=1}^T\sum_{i=1}^K \mathbb{E}[\ell_{t,i}]$ instead of $\eta TK$. We expect that the log-barrier based LB-Prod is a better candidate for second-order bounds or path bounds such as the ones derived in Wei and Luo 2018. or Ito 2021 as the algorithm does not introduce any perturbations.
>
> We thank the reviewer for pointing out the typos and plan to implement the minor comment fixes as part of the final version of the paper.
>
> Allenberg, Chamy, et al. "Hannan consistency in on-line learning in case of unbounded losses under partial monitoring." Algorithmic Learning Theory: 17th International Conference, ALT 2006, Barcelona, Spain, October 7-10, 2006. Proceedings 17. Springer Berlin Heidelberg, 2006.
>
> Ito, Shinji. "Parameter-free multi-armed bandit algorithms with hybrid data-dependent regret bounds." Conference on Learning Theory. PMLR, 2021.
>
> Wei, Chen-Yu, and Haipeng Luo. "More adaptive algorithms for adversarial bandits." Conference On Learning Theory. PMLR, 2018.

---

> > ### Comment · Reviewer_rzG8 · 2024-08-14
> >
> > I thank the authors for answering my questions. I keep my original positive score.

---

### Official Review · Reviewer_Noj5 · 2024-07-13

**Soundness:** 3
**Presentation:** 3
**Contribution:** 2
**Rating:** 5
**Confidence:** 4

**Summary:**

The paper studies the multi-armed bandit (MAB) problem. The main focus is on the Prod algorithm, which is fundamentally considered to be sub-optimal for MAB settings. The authors challenge this conjecture by leveraging Prod's interpretation as a first-order Online Mirror Descent (OMD) approximation. They make following main contributions: first, they present variants of Prod that achieve optimal regret bounds for adversarial MABs. Second, they propose an algorithm that achieves best-of-both-worlds guarantees, showing logarithmic regret in the stochastic bandit setting. The authors emphasize the simplicity and efficiency of their approach, using arithmetic update rules instead of solving complex optimization problems. The paper systematically develops these ideas, starting from the problem definition and related work, moving through the theoretical analysis of the proposed modifications, and proof of their performance guarantees.

**Strengths:**

The paper has following strengths:

- The paper is well written and easy to understand
- Authors provide a very good overview of the current literature related to the work presented in this paper
- The paper challenges and disproves a long-standing conjecture about the sub-optimality of Prod in the context of MAB
- Theoretical results appear to be correct

**Weaknesses:**

- One main weakness of the paper is that authors do not provide any experimental results. I highly encourage authors to provide experimental results to illustrate their theoretical claims

- Another main weakness is that authors do not provide any comparison (discussion or results) with other algorithms that achieve optimal regret guarantees in similar bandit settings

- Authors do not provide any discussion on real world applications of their algorithms

- The focus of the paper is heavily theoretical, which might reduce its accessibility and usability for practitioners looking for direct application insights.

**Questions:**

Can you provide experimental results to illustrate the theoretical claims?

How does your algorithms compare with other algorithms in similar settings?

**Limitations:**

While the theoretical contributions are strong, the lack of empirical validation means that the practical effectiveness of the algorithms in environments remains to be seen.

The scope of the paper is restricted to multi-armed bandits, and the paper does not discuss the potential application of the proposed methods to other types of online learning problems.

---

> ### Author Rebuttal · Authors · 2024-08-06
>
> We address weaknesses and questions below:
>
> Weaknesses:
>
> --No experimental results: This is a purely theoretical paper. Our goal was to develop new variants of Prod which enjoy min-max optimal regret in the adversarial setting and instance-dependent optimal regret bounds in the stochastic setting.
>
> --Comparison with other algorithms: We are happy to include a clear discussion explaining that the algorithms are expected to perform very similarly to their FTRL/OMD counterparts (as seen from the regret bounds and the perturbation argument) but with the added benefit of having a closed form.
>
> --Real world application: The MAB problem is very well studied and there is a vast amount of literature on this topic which describes many real world applications. Our work proposes new, arguably simpler, algorithms for solving the MAB problem and as such has the same real world applications as prior MAB algorithms. For a good review of the MAB problem we point the reviewer to the Bandit Algorithms by Tor Lattimore and Csaba Szepesvari and Regret Analysis of Stochastic and Nonstochastic Multi-armed Bandit Problems by Sébastien Bubeck, Nicolò Cesa-Bianchi
>
> --The focus of the paper is heavily theoretical:
> While our paper is indeed heavy in theory, we do not expect that it is less accessible to practitioners as our proposed algorithms are described well and easy to implement. We are happy to include complete pseudo-code in the appendix for our proposed algorithms. Further we plan to implement the suggestions of reviewer **r5go** to further improve the presentation
>
>
> Questions:
> >Can you provide experimental results to illustrate the theoretical claims?
>
> At this time we do not plan on conducting an empirical evaluation of our techniques.
>
> >How does your algorithms compare with other algorithms in similar settings?
>
> See our comment regarding comparison with other algorithms.

---

> > ### Comment · Reviewer_Noj5 · 2024-08-07
> >
> > Thank you for the response. I will keep my score.

---

### Author Rebuttal · Authors · 2024-08-06

We would like to thank all the reviewers for their careful reviews and suggestions on how to improve our work. We address each of the comments regarding weaknesses and each of the questions separately under the respective reviewer.

---

### Decision · Program_Chairs · 2024-09-25

**Decision:**

Accept (poster)

**Comment:**

This paper considers the bandit problem, specifically the Prod algorithm, which updates a constant share based on the loss function. Due to its linearity with respect to the loss function, it is incentive-compatible (e.g., [Freeman et al. 2020]) in the sense that minimizing the expected loss maximizes the weight in the next round. An existing algorithm called WSU-UX has a $T^{2/3}$ regret bound in the adversarial case, but this paper shows that a modified loss function can eventually lead to a $\sqrt{KT}$ bound as well as a rate-optimal stochastic bandit bound. All reviewers are positive about the paper. I recommend that the authors prepare the camera-ready version that incorporates reviewers' suggestions.